# The bulk parameterizations of turbulent air-sea fluxes in NEMO4: the origin of Sea Surface Temperature differences in a global model study

Giulia Bonino[1*], Doroteaciro Iovino[1], Laurent Brodeau[2], and Simona Masina[1]

[1]Ocean Modeling and Data Assimilation Division, Centro Euro-Mediterraneo sui Cambiamenti Climatici, Bologna, Italy.
[2]Centre National de la Recherche Scientifique, IGE/MEOM, Grenoble, France

**Correspondence:** Giulia Bonino (giulia.bonino@cmcc.it)

**Abstract.**

Wind stress and turbulent heat fluxes are the major driving forces which modify the ocean dynamics and thermodynamics. In the Nucleus for European Modelling of the Ocean (NEMO) ocean general circulation model, these turbulent air-sea fluxes (TASFs) can critically impact the simulated ocean characteristics. This paper investigates how the different bulk parameterizations to calculate turbulent air-sea fluxes in NEMOv4 drives substantial differences in sea surface temperature (SST). Specifically, we study the contribution of different aspects and assumptions of the bulk parameterizations in driving the SST differences in NEMO global model configuration at ¼ degree of horizontal resolution. These include the use of the skin temperature instead of the bulk SST in the computation of turbulent heat flux components and the estimation of wind stress and of turbulent heat flux components which vary in each parameterization due to different bulk transfer coefficients. The analysis of a set of short-term sensitivity experiments, where the only change is related to one of the aspects of the bulk parameterizations, shows that parameterization-related SST differences are primarily sensitive to the wind stress differences and to the implementation of skin temperature in the computation of turbulent heat flux components. Moreover, in order to highlight the role of SST-turbulent heat flux negative feedback at play in ocean simulations, we compare the TASFs differences obtained using NEMO ocean model with the estimations from Brodeau et al. (2017), who compared the different bulk parameterizations using prescribed SST. Our estimations of turbulent heat flux differences between bulk parameterizations is weaker than that found by Brodeau et al. (2017).

## 1 Introduction

Ocean and atmosphere circulations are highly influenced by the transfer of momentum and heat at the air-sea interface (e.g., Gill, 1982; Siedler et al., 2013). These transfers of energy are primarily driven by surface radiative flux and turbulent air–sea fluxes (TASFs), which include wind stress and the turbulent heat flux components (THFs, latent and sensible heat fluxes). In the upper ocean, the wind stress is a major driving force for basin-scale circulation (e.g., Chen et al., 1994; Shriver and Hurlburt, 1997), and the THFs are important for determining its thermal properties (e.g., Yuen et al., 1992; Swenson and Hansen, 1999).

Therefore, both wind stress and THFs are important for the evolution of sea surface temperature (SST), because of their contribution to turbulent mixing within the ocean surface mixed layer (e.g., Barnier, 1998).

Since direct observations of TASFs are sparse in space and time, the estimates of TASFs are derived using bulk formulas, which relate each component of turbulent air-sea flux to more easily measurable and widely available meteorological surface atmospheric variables (e.g. wind speed, air temperature, air specific humidity) through bulk transfer coefficient. These bulk transfer coefficients are estimated using bulk parameterizations. Different bulk parameterizations are currently used and they are traditionally developed statistically, comparing in situ meteorological observations of surface atmospheric variables with

TASFs derived from ship and buoy measurements (Large and Pond, 1981, 1982; Smith, 1988; Fairall et al., 1996, 2003; Bradley and Fairall, 2007; Edson et al., 2013).

In NEMO ocean general circulation model (OGCM), TASFs are computed by means of bulk formulas using prescribed surface atmospheric variables (air temperature, air humidity, wind) and the prognostic SST of the model (hereinafter online prognostic SST approach). The online prognostic SST approach allows to incorporate the response of the ocean (i.e. SST) to

atmospheric events into the estimation of the THFs and of longwave radiation (i.e. non solar heat flux components, NSHFs) at every time step of the numerical experiment. The possibility of feedback mechanisms between the ocean and the atmosphere partially simulates the energy exchange between them (Kara et al., 2000). The approach requires the choice of a given bulk parameterization, which influences the magnitude of the wind stress and of the THFs (Kara et al., 2000). The TASFs affect the simulated ocean characteristics and in particular the evolution of the SST (Torres et al., 2019). It is worth mentioning that the

online prognostic approach does only partially close the air-sea feedback. Surface winds and clouds are affected by the SST structure on daily time-scale which, in turn, affect the SST and the TASFs (Desbiolles et al., 2021; de Szoeke et al., 2021; Gaube et al., 2019; Li and Carbone, 2012; Small et al., 2008). The closed air-sea feedback (hereinafter coupled approach) in the system might substantially impact the turbulent fluxes (Lemarié et al., 2021; Small et al., 2008), but the coupled approach is still not yet mature in the ocean model community. Recently Lemarié et al. (2021) implemented a first attempt of a simplified

atmospheric boundary layer model (ABL) to improve the representation of air-sea interactions in NEMOv4.2. However, the online prognostic SST approach is still largely used by the ocean modeling community in a variety of applications.

Brodeau et al. (2017) compared a set of bulk parameterizations which compute TASFs using prescribed SST (hereinafter offline prescribed SST approach) rather than prognostic SST of the model. Based on their approach Brodeau et al. (2017) reported that the use of different bulk parameterizations to estimate TASFs can typically produce differences in total turbu-

lent heat flux ($Q_T$, i.e. the sum of the THFs, latent and sensible heat fluxes) of about $10W/m^2$ and in wind stress of about $20mN/m^2$. The online prognostic SST approach, used by the NEMO experiments performed for this study, can substantially modify these estimations through the negative SST feedback on $Q_T$, likely dampening the $Q_T$ discrepancies across the different bulk parameterizations (Seager et al., 1995).

The purpose of this work is to better understand the response of the prognostic SST to the TASFs and to their parameteriza-

tion in NEMO version 4.0 at $1/4°$ of horizontal resolution, and to discuss the role of the SST-$Q_T$ negative feedback at play in the online prognostic SST approach. We address the sensitivity of the SST to various aspects of the different bulk parameterizations such as the inclusion of the skin temperature in the computation of the THFs and the role of the bulk transfer coefficients

in the estimation of the wind stress and the THFs. In order to do that, we analysed differences between short-term sensitivity experiments where bulk assumptions are excluded (e.g skin temperature) or bulk transfer coefficients are computed mixing the different bulk parameterizations. Lastly, in order to highlight the role of the SST-$Q_T$ negative feedback at play in our online prognostic SST approach, we compare the TASFs with the estimations from Brodeau et al. (2017). We also provide a simple validation of the different experiments against a SST observed dataset, but the main objective of the work is to investigate the impact of a set of bulk parameterizations on the SST generated by NEMO rather than evaluate their accuracy in reproducing it.

This paper is organized as follows: in section 2, we present the model used for this study, a short overview of the bulk formulas implemented in NEMOv4, the experimental set-up and the modifications introduced in the bulk parameterizations to performed sensitivity experiments. In section 3 we present the parameterization-related SST discrepancies, we quantify SST discrepancies related to various aspects of the different bulk parameterizations and we compare and discuss our finding in relation to existing works. Our conclusions are summarized in section 4.

## 2   Model configuration, bulk forcing and experimental set-up

### 2.1   NEMOv4 model configuration

The sensitivity of prognostic SST to bulk parameterizations is investigated in a numerical study using the Nucleus for European Modelling of the Ocean[1] (NEMO, version 4.0, revision 12957). NEMO is a three-dimensional, free-surface, hydrostatic, primitive-equation global ocean general circulation model (Madec G. and NEMO System Team, ) coupled to the Sea Ice modelling Integrated Initiative (SI[3], NEMO Sea Ice Working Group, 2020). Our configuration uses the global ORCA025 tripolar grid (Madec and Imbard, 1996) with 1/4° horizontal resolution ( 27.75km) at the Equator, which increases with latitudes, e.g. 14km at 60 °. The vertical grid has 75 levels, whose spacing increases with a double hyperbolic tangent function of depth from 1 m near the surface to 200 m at the bottom, with partial steps representing the bottom topography (Bernard et al., 2006). The model bathymetry is based on the combination of ETOPO1 data set (Amante and Eakins, 2009) in the open ocean and GEBCO (IOC, 2003) in coastal regions. The horizontal turbulent viscosity is parameterized by means of a biharmonic function with a value of $1.8 \times 10^{11} m^4 s^{-1}$ at the Equator, reducing poleward as the cube of the maximum grid cell size. The advection of the tracers uses a total variance dissipation (TVD) scheme (Zalesak, 1979). The Laplacian lateral tracer mixing is along isoneutral surfaces with a coefficient of $300 m^2 s^{-1}$. The vertical mixing of tracers and momentum is parameterised using the turbulent kinetic energy (TKE) scheme (Blanke and Delecluse, 1993). Subgrid-scale vertical mixing processes are represented by a background vertical eddy diffusivity of $1.2 \times 10^{-5} m^2 s^{-1}$ and a globally constant background viscosity of $1.2 \times 10^{-4} m^2 s^{-1}$. The bottom friction is quadratic and a diffusive bottom boundary layer scheme is included. The continental runoff data are a monthly climatology derived from the global river flow and continental discharge data set for the major rivers (Dai and Trenberth, 2002; Dai et al., 2009), and estimates by Jacobs et al. (1996) for the Antarctic coastal freshwater discharge. The initial

---

[1]https://www.nemo-ocean.eu/

conditions for temperature and salinity are provided by World Ocean Atlas 2013 (Levitus et al., 2013). All the experiments are forced with the hourly ERA5 Reanalysis of the ECWMWF (Hersbach et al., 2020).

## 2.2   The bulk formulas and their parameterization in NEMO4.0

As stated in the introduction, NEMO uses the online prognostic SST approach to compute TASFs, which are estimated using the prognostic SST and prescribed atmospheric surface variables by means of aerodynamic bulk formulas:

$$\tau = \rho C_D u \mathbf{u_z} \tag{1a}$$

$$Q_H = \rho C_p C_H (\theta_z - T_s) U \tag{1b}$$

$$E = \rho C_E (q_0 - q_z) U \tag{1c}$$

$$Q_L = -L_v E \tag{1d}$$

where $\tau$ is the wind stress, $Q_H$ is the turbulent flux of sensible heat , $E$ is the evaporation, and $Q_L$ is the turbulent flux of latent heat. Throughout this paper, we use the convention that a positive sign of $\tau$, of THFs ($Q_H$ and $Q_L$), and of the total turbulent heat flux $Q_T$ ($Q_T = Q_H + Q_L$) means a gain of the relevant quantity for the ocean. The term $\rho$ is the density of air; $C_p$ is the heat capacity of moist air, and $L_v$ is the latent heat of vaporization. $\mathbf{u_z}$ is the wind speed vector at height z, which may be referred to the ocean currents. The bulk scalar wind speed $U$ is the scalar wind speed $|\mathbf{u_z}|$ with the potential inclusion

of a gustiness contribution. The convective gustiness is a temporary increase of the wind speed due to the friction and the free convection and it is active and significant in very calm wind conditions with unstable near-surface atmosphere. It is added to the wind speed and it avoids zero wind singularity. $\theta_z$ and $q_z$ are the potential temperature and specific humidity of air at height z, while $T_s$, $q_0$ are he potential temperature and specific humidity at surface. Depending on the bulk parameterization used, $T_s$ can be the temperature at the air-sea interface (sea surface skin temperature, SSTskin) or at typically 1 meter deep (bulk sea

surface temperature, SST). The SSTskin differs from the SST due to the contributions of two effects of opposite sign: the cool skin and warm layer (CSWL). The cool skin is the cooling of the millimeter-scale uppermost layer of the ocean to ensure a steep vertical gradient of temperature which sustains the heat flux continuity between ocean and atmosphere. The warm layer is the warming of the upper few meters of the ocean under day and sunny conditions.

$C_D$, $C_H$, and $C_E$ are the Bulk Transfer Coefficients (BTCs) for wind stress, sensible heat, and moisture, respectively.

Therefore, the main differences among bulk parameterizations are usually related to:

1. The use of the skin temperature (hereinafter SSTskin) rather than the bulk SST in the estimation of near surface atmospheric stability and bulk formulas.

2. The form of the exchange coefficients

3. The inclusion of convective gustiness in wind calculation

4. The effect of including ocean current in stress

In this study, we attempt to disentangle the effects of the first two aspects on SST (section 3.2,3.3 and 3.4), and we discuss the effect of the inclusion of convective gustiness in the wind stress computation (section 3.4). The effect of the ocean current interaction/feedback in the bulk formulation has been widely explored in the literature (e.g. Renault et al., 2019a, b; Sun et al., 2019). Although many previous studies highlighted the substantial difference in the surface input to the ocean between calculations that use absolute vs. relative wind, we have preferred to leave this aspect to further work since the implementation of this correction does substantially depend on the characteristics of the forcing fields (Renault et al., 2020).

The online prognostic SST approach of NEMO uses the modelled SST at each time step to estimate NSHFs (i.e. THFs + long wave radiation). In our experiments, we only focus on the NSHFs computed by bulk formulas, namely the THFs. The SST is responding to the total turbulent heat flux $Q_T$ at each time step: the $Q_T$ generate SST anomalies, and SST anomalies, in turn, can modulate $Q_T$. Specifically, SST and $Q_T$ feedback negatively: when the SST gets anomalously cold, then $Q_T$ increases, and that means that $Q_T$ increases in response, the SST will tend to increase and the $Q_T$ to decrease and so on. This negative feedback of the online prognostic SST works to reduce the heat fluxes difference across the different bulk parameterizations. On the other hand, the wind stress is not affected by the this type of first-order feedback at play for the $Q_T$.

In this study we focus on three of bulk parameterizations implemented in NEMOv4: NCAR (Large and Yeager, 2009), COARE 3.6 (Edson et al. (2013) + Chris Fairall, private communication, hereinafter referred to as "COARE"), and ECMWF as coded in the Aereobulk package (Brodeau et al., 2017). All the codes to estimate TASFs in the NEMOv4.0 framework, originates from this AeroBulk package, which is completely open source and available at https://github.com/brodeau/aerobulk (Brodeau et al., 2017).

COARE and ECMWF parameterizations are meant to be used with the SSTskin, so that the two algorithms include a CSWL parameterization to estimate SSTskin. NCAR uses the bulk SST in heat fluxes calculation and the zero wind singularity is avoided by simply setting a minimum value for the scalar wind speed to $0.5 m/s$. To calculate the BTCs, the bulk parameterizations rely on an empirical closure. More specifically, in COARE and ECMWF parameterizations, the computation of BTCs relies on the Monin-Obukhov similarity theory (MOST, Monin and Obukhov, 1954). As such, BTCs are function of the roughness lengths and of the stability of the atmospheric surface layer. The NCAR parameterization uses a combination of the MOST theory with a semi-empirical form of drag coefficient in which the BTCs are computed as function of neutral wind speed (e.g. the wind speed at neutral stability condition and at 10m reference level, $U_{N10}$).Then, the BTCs are shifted to the current atmospheric stability. Figure 1 shows the $U_{N10}$ annual mean and the neutral BTCs as a function of $U_{N10}$ for the selected bulk formula parameterizations. Due to the stronger neutral drag coefficient $C_D^{N10}$, NCAR parameterization tends

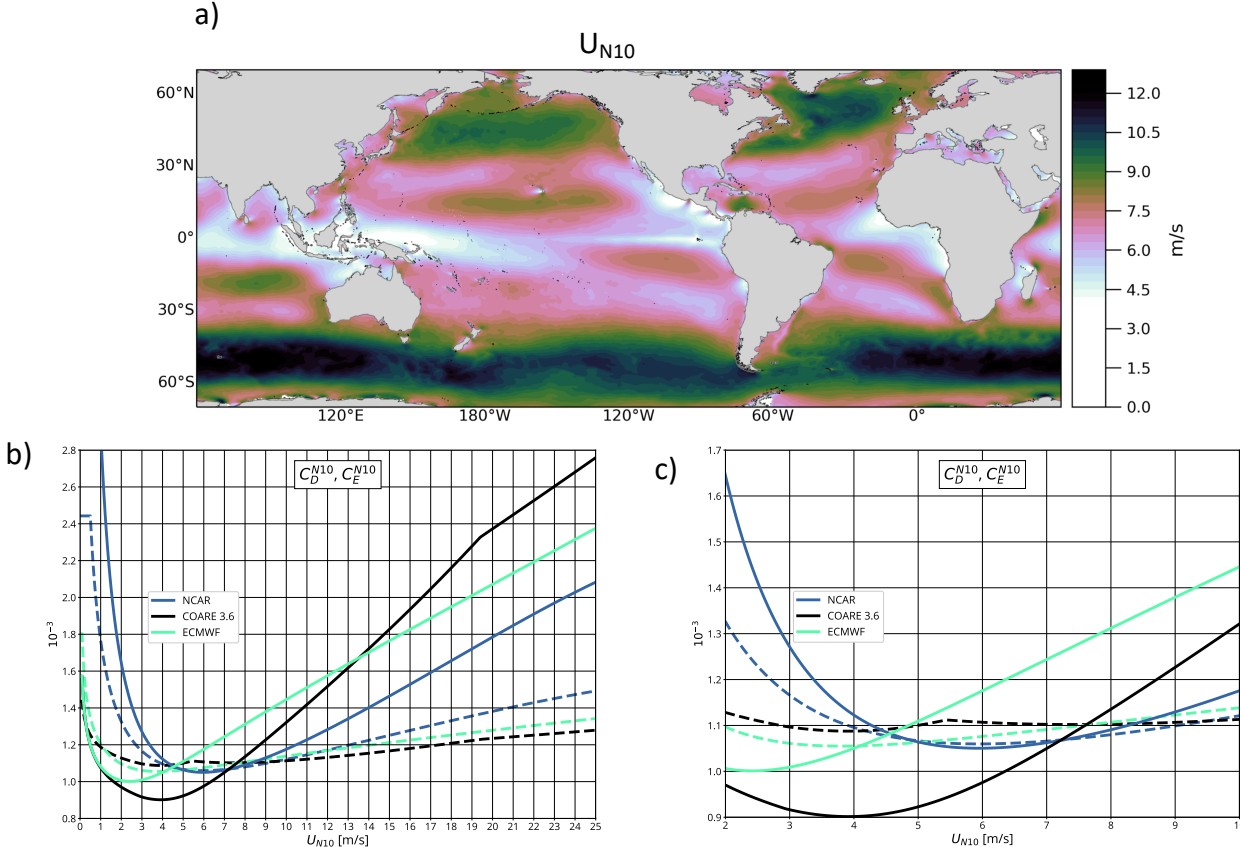

**Figure 1.** a) Annual mean of $U_{N10}$ from NCAR parameterization b) Neutral drag and moisture transfer coefficients ($C_D^{N10}$ and $C_E^{N10}$) for COARE (black), NCAR (blue), and ECMWF (green) bulk parameterizations (solid and dashed lines, respectively), as functions of the neutral wind speed at 10 m; c) zoom of pannel b) for the wind range $2-10m/s$

to enhance wind stress with respect to COARE and to lower extend to ECMWF under light wind condition ($u < 5m/s$). On the other hand, ECMWF parameterization enhances wind stress with respect to NCAR and COARE for wind speed above $5m/s$, while COARE enhances it for wind speed above $13m/s$. For the discussion of the following results, it is important to highlight the wind speed range where the NCAR $C_D^{N10}$ function intersects ECMWF and COARE $C_D^{N10}$ functions (Figure 1c). In the range of 7-9 m/s, the $C_D^{N10}$ of COARE is smaller than ECMWF, but slightly higher or approximately equal (around 7 m/s) than NCAR $C_D^{N10}$. In the range of 4-5 m/s, the $C_D^{N10}$ of ECMWF is slightly smaller or approximately equal than NCAR, but higher than COARE $C_D^{N10}$. Under weak conditions, NCAR parameterization tends to enhance evaporation with respect to COARE and ECMWF due to the stronger $C_E^{N10}$ (see Figure 1). For detailed explanation of BTCs derivation for each bulk parameterization please refer to the technical report by Bonino et al. (2020).

## 2.3 Experimental set-up

In order to investigate the role of different aspects of bulk parameterizations in driving prognostic SST, we performed six
numerical experiments (Table 1). All the experiments are 1-year long experiments, starting from January 2016 after 1-year of
spinup. There is no intent to analyze this year in relation to a specific climatic mode. The simulation are forced by the hourly
surface atmospheric variables of the ERA5 Reanalysis (Hersbach et al., 2020).

We first performed three experiments (hereinafter 'control experiments') in order to quantify the bulk parameterization-related
SST discrepancies. In particular, we performed `ECMWF_S`, `COARE_S` and `NCAR` experiments, which use the ECMWF, COARE
and NCAR parameterizations, respectively. `ECMWF_S` and `COARE_S` experiments use the SSTskin (through their respective
CSWL scheme) and consider convective gustiness in wind speed calculation. As opposed, `NCAR` experiment computes THFs
using bulk SST and the convective gustiness is not considered in wind speed computation.

In order to disentangle the contribution of the skin temperature and the contribution of the different wind stress and THFs
in driving sea surface temperature differences, we performed two sensitivity experiments (hereinafter 'mixed experiments').
First, we performed `ECMWF_NS` experiment, which uses the ECMWF parameterization, and THFs are computed using bulk
SST rather than SSTskin. Second, we run `CdNC_CeEC_NS` experiment, which uses the ECMWF parameterization to calculate
$C_H$ and $C_E$ BTCs and the NCAR bulk formula to calculate $C_D$ BTC. THFs are computed using bulk SST. Moreover, we
performed an additional experiment, called `ECMWF_NS_NG`, which differs from `ECMWF_NS` only for the exclusion of the
convective gustiness in the wind speed calculation.

The comparison between `ECMWF_S` and `ECMWF_NS` is used to determine the Skin Temperature contribution in driving THFs
differences and in turn SST differences. The comparison between `CdNC_CeEC_NS` and `ECMWF_NS`, which differ only for the
$C_D$ BTC computation, and between `CdNC_CeEC_NS` and `NCAR`, which differ only for the $C_H$ and $C_E$ BTCs computation,
teach us about the wind stress and the THFs differences contribution in driving SST differences, respectively. Moreover, we
compare `ECMWF_NS_NG` and `ECMWF_NS` experiments to show the effect of the inclusion of convective gustiness in the wind
speed calculation on wind stress computation (shown in the supplementary material). We analyze annual mean differences
between experiments. We use the absolute wind, e.g. the parameterizations do not include the ocean currents feedback to
calculate wind in equation 1a.

| Experiment name | sea surface temperature used ($T_s$) | computation of $C_D$ | computation of $C_E$ and $C_H$ | convective gustiness |
|---|---|---|---|---|
| COARE_S | SSTskin | COARE3.6 | COARE3.6 | Yes |
| ECMWF_S | SSTskin | ECMWF | ECMWF | Yes |
| NCAR | SST | NCAR | NCAR | No |
| ECMWF_NS | SST | ECMWF | ECMWF | Yes |
| CdNC_CeEC_NS | SST | NCAR | ECMWF | No |
| ECMWF_NS_NG | SST | ECMWF | ECMWF | No |

**Table 1.** Summary of the numerical experiments.

## 3 Results

We here discuss the parameterization-related discrepancies in terms of TASFs, SST and meridional heat transport (section 3.1), and we describe the sensitivity of the prognostic SST of the model to various aspects of the bulk parameterizations (section 3.2, 3.3). Except for sections 3.1 and 3.2, we consider only experiments which estimate the NSHFs using $T_s = SST$ in order to disentangle the THFs and wind stress differences contribution to the prognostic SST without the effect of the CSWL implementation.

### 3.1 Parameterization-related discrepancies

We compare the SST simulated by the ECMWF_S, COARE_S and NCAR control experiments with the European Space Agency (ESA) Climate Change Initiative (CCI) SST dataset v2.0 (hereinafter ESA CCI SST dataset) which consists of daily-averaged global maps of SST on a 0.05° x 0.05° regular grid, covering the period from September 1981 to December 2016 (Merchant et al., 2019). All the control experiments present a warm bias in the Eastern Pacific, in the Eastern Boundary Upwelling systems (EBUS), in the Western Boundary Currents (WBCs) and in the Antarctic Circumpolar Current (ACC) region. The SST reproduced by COARE_S and ECMWF_S shows a cold bias of about -1°C in the North Atlantic open ocean at mid-latitudes, and a warm bias of about 0.5°C in the Indian Ocean and the Western Pacific (Figure 2a,b); NCAR SST is also colder than observations, with a larger bias of about -2°Cin the North Atlantic (Figure 2c). The bias is generally higher compared with other two experiments and covers wider areas.

Figure 3 shows the differences in total turbulent heat fluxes, wind stress and wind stress curl, from ECMWF_S and COARE_S with respect to NCAR. ECMWF_S wind stress is slightly weaker with respect to NCAR over the equatorial band and it is stronger elsewhere (Figure 3a). In COARE_S the wind stress is weaker than NCAR over a broader region with respect to ECMWF_S, namely over the areas characterized by calm wind conditions (see Figure 1). The wind stress curl ($WSC$) patterns are similar for the two pairs of differences (Figure 3c), they differ only for their magnitude. As regards the $Q_T$ differences (Figure 3b), a gain of heat for ECMWF_S is a clear feature over the Pacific and Atlantic equatorial regions and over EBUS with respect to NCAR.

These TASFs likely drive substantial SST differences between experiments (Figure 4). While the SST in COARE_S is warmer than in NCAR everywhere, the SST in ECMWF_S is overall warmer than in NCAR, but with a colder area (down to -0.6°C) over EBUS and over Pacific and Atlantic equatorial regions. This spatial pattern of SST differences persists when extending the simulations up to 5 years (not shown). In these experiments, which differ only in the bulk parameterization, the SST differences can arise from the differences in the wind stress and in the THFs as computed by the chosen bulk parameterization (Figure 3). In particular, the wind stress discrepancies, due to the computation of $C_D$ and to the inclusion of the convective gustiness, may impact on the ocean dynamics by modifying the 3D ocean circulation and hence the pattern of the SST. The differences in THFs, due to the $C_E$ and $C_H$ computation and to the cool-skin/warm layer CSWL scheme, may affect the SST through modification of the heat loss to the atmosphere (dominated by evaporation in this region). Furthermore, differences in the wind stress and in THFs may also act together by amplifying or damping their single effect on the SST.

Changes on the simulated SST can reflect on the temperature profile in the upper ocean and the distribution of heat on global scales. We have computed the global ocean heat transport in the upper 100 meters and compared it among experiments. Figure 4 (c,d) presents the meridional heat transport (MHT) as a function of latitude. The MHT is larger in ECMWF_S compared to NCAR mostly at all latitudes (Figure 4c), with the largest differences (about 0.8 PW, 20% of NCAR absolute value) in the tropical band where ECMWF_S wind stress is stronger than NCAR one (Figure 3a). COARE_S and NCAR compare well, with differences lower than 0.3 PW (Figure 4d). Then, we will focus only on the differences between ECMWF_S andNCAR to analyze in detail the relationship between TASFs and SST. We show differences in MHT only when relevant.

## 3.2 Skin temperature

The ECMWF and COARE parameterizations, in contrast to NCAR, expect SSTskin as the surface temperature input in order to estimate the near surface atmospheric stability and to compute the THFs. The SSTskin is also used to estimate the upward long wave flux, needed by the CSWL scheme as component of the NSHFs. Here, we compare the results between ECMWF_S and ECMWF_NS to understand the impact of the CSWL implementation in driving the differences in the THFs and by consequence in the SST shown in Figure 4 (see Table 1 for experiments details). We discuss the impact of the use of skin temperature for ECMWF parameterization, but similar results are found using COARE (not shown). The ECMWF_S experiment uses the CSWL scheme, so that $T_s \equiv SSTskin$ is used to compute THFs, as opposed to ECMWF_NS in which $T_s \equiv SST$. Consideration of the CSWL effect yields a SST global mean warming of 0.2°C (Figure 5c), with a maximum of 0.3°C over the western equatorial Pacific Ocean, in the Indo-Pacific Warm Pool. In the tropical eastern and Northern Pacific Ocean, and over ACC, the differences

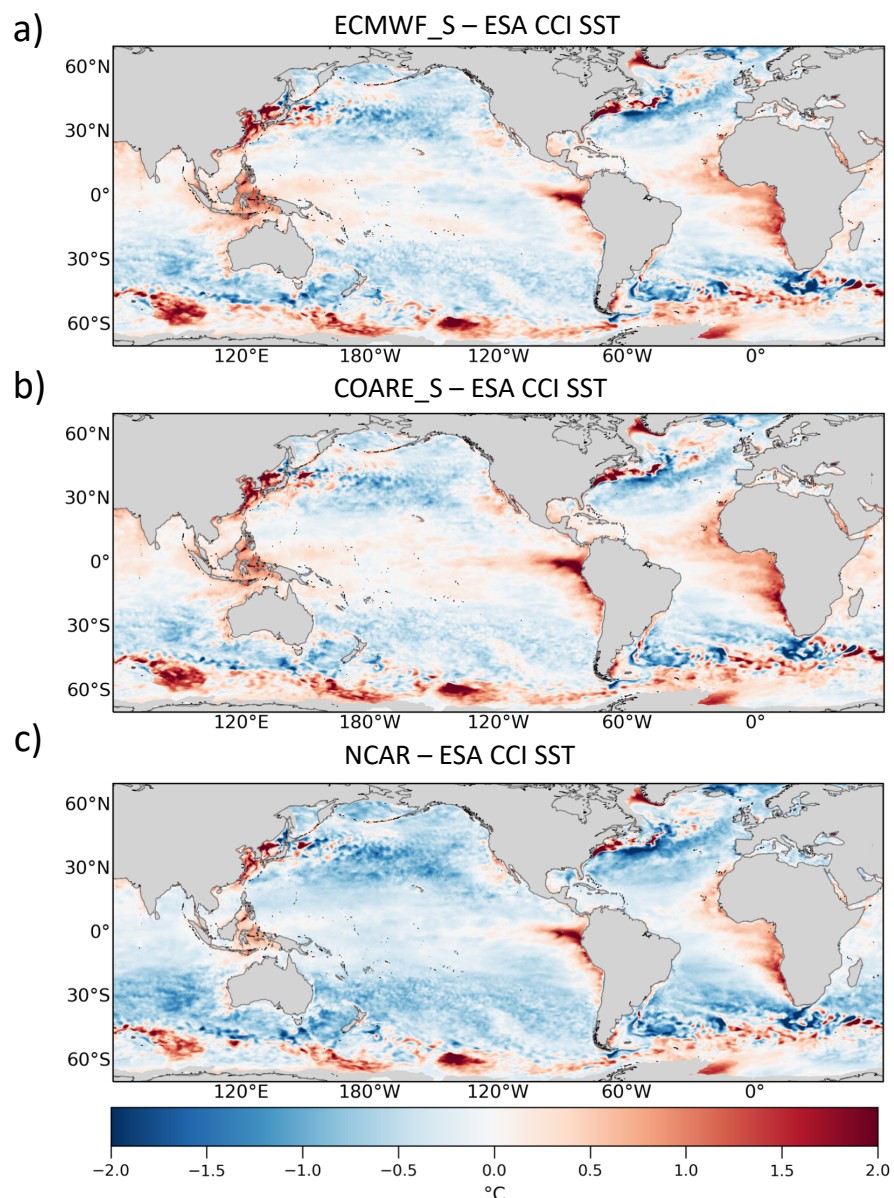

**Figure 2.** Annual mean SST differences between a) `ECMWF_S` b) `COARE_S`, c) `NCAR` against ESA CCI SST.

are below 0.1°C. The global-mean SSTskin tends to be about 0.1°C colder than the SST (Figure 5a). On a global average basis, the cool skin process dominates over to the warm layer effect. Specifically, evaporation occurs almost everywhere and most of the time, while the warm layer builds up under sunny and low wind conditions.

The colder $T_s$ in `ECMWF_S` with respect to `ECMWF_NS` yields a slightly weaker heat loss to the atmosphere due to the decreased NSHFs (mostly evaporation). In `ECMWF_S` the weaker heat loss to the atmosphere implies a gain of heat by the ocean (positive

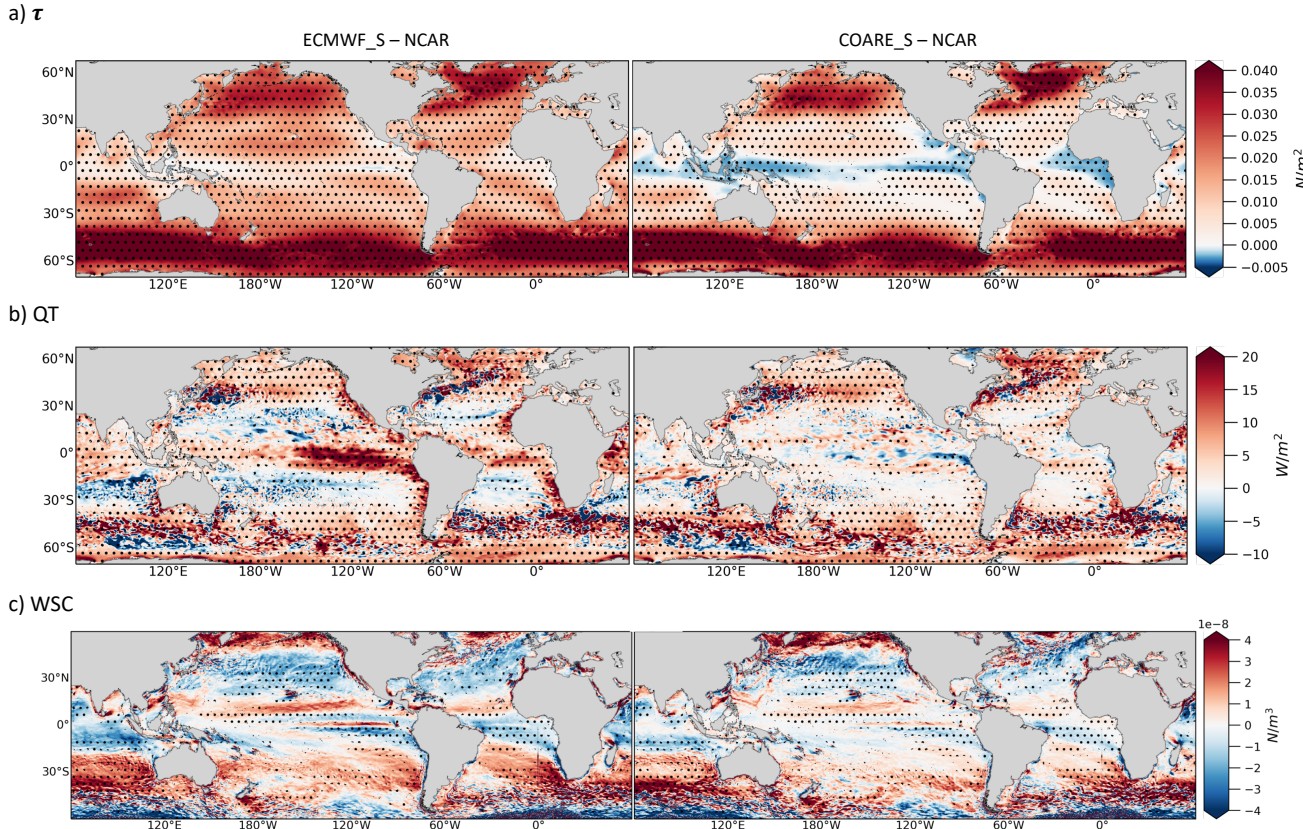

**Figure 3.** Annual mean differences between experiments of a) wind stress ($\tau$) and b) total turbulent heat fluxes ($QT$) and c) wind stress curl ($WSC$) between `ECMWF_S` and `NCAR` experiments (left) and `COARE_S` and `NCAR` experiments (right). Hatching indicates significant values (95% confidence level)

regions in Figure 5b) of approximately $1W/m^2$ on global average compared to `ECMWF_NS`. We can conclude that the negative SST discrepancies between parameterizations noted in Section 3.1 (Figure 4a) are not explained by the use of the CSWL scheme in the ECMWF parameterization. The SST differences between `ECMWF_NS` and `NCAR` (Figure 6a) with respect to the SST differences between `ECMWF_S` and `NCAR` (Figure 4) present a reduction of the overall warm temperature differences, but

245     the cold temperature difference over the tropical Pacific and Atlantic and over the EBUS are still present.

### 3.3   Turbulent Heat fluxes

In order to investigate the effect of the different computation of the THFs between `ECMWF_S` and `NCAR` in driving SST differences (Figure 4a), we compare the results between `CdNC_CeEC_NS` and `NCAR` (see Table 1 for experiments details).

    The SST differences between `CdNC_CeEC_NS` and `NCAR` does not show the cold bias over EBUS and over equatorial

250     Atlantic and Pacific as we found for between experiments `ECMWF_S` and `NCAR` (compare Figure 4a with Figure 6c). Over

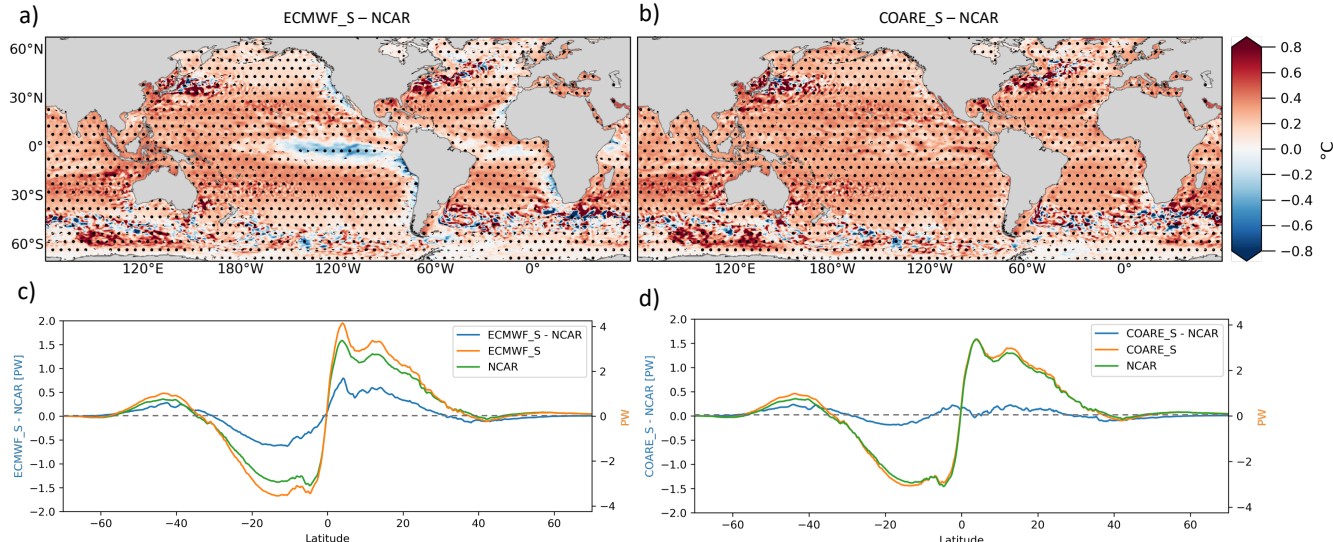

**Figure 4.** Annual mean SST differences between a) `ECMWF_S-NCAR` and b) `COARE_S - NCAR`; Global Meridional Heat Transport values on the right y axis) and differences (values on the left y axis) in the upper 100m ocean between c) `ECMWF_S` and `NCAR` and d) `COARE_S` and `NCAR`. Hatching indicates significant values (95% confidence level).

those areas, the SST in `CdNC_CeEC_NS` is warmer than in `NCAR` of about $0.3°C$ on average.

As shown in Figure 7a, `CdNC_CeEC_NS` receives an excess of $Q_T$ of about $1W/m^2$ on average with respect to `NCAR`. The main contributor to this difference is the latent heat (Figure 7a and Figure 8b), resulting from the use of a different $C_E$ in the two experiments. Indeed, the $C_E$ of `CdNC_CeEC_NS`, which is smaller than $C_E$ of `NCAR` (Figure 8a), induces weak evaporation. The resulting weaker heat loss to the atmosphere in `CdNC_CeEC_NS` with respect to `NCAR` implies a gain of heat by the ocean (positive regions in Figure 7a) of about $2W/m^2$ over low-latitudes and up to $6\ W/m^2$ over mid-latitudes (Figure 7b). A similar process is acting also in areas where the annual mean pattern of $QT$ is patchy due to the mesoscale activities in both in summer and winter seasons (e.g. in the Western Boundary Currents, Figure S1). In `CdNC_CeEC_NS`, the negative virtual temperature differences at the air-sea interface are smaller than `NCAR`, inducing weaker heat loss from the ocean to the atmosphere.

The differences in $Q_T$ and SST have the same sign, which suggests that the $Q_T$ drive the SST differences. As it is clearly shown by the annual zonal-mean differences time-series (Figure 7b): the weaker the heat loss from the ocean in `CdNC_CeEC_NS` along the latitude, the warmer the ocean modeled by `CdNC_CeEC_NS` experiment with respect to `NCAR`.

In summary, weak evaporation and, by consequence, the weaker heat loss in `CdNC_CeEC_NS` generates an ocean surface temperature that is warmer than `NCAR`.

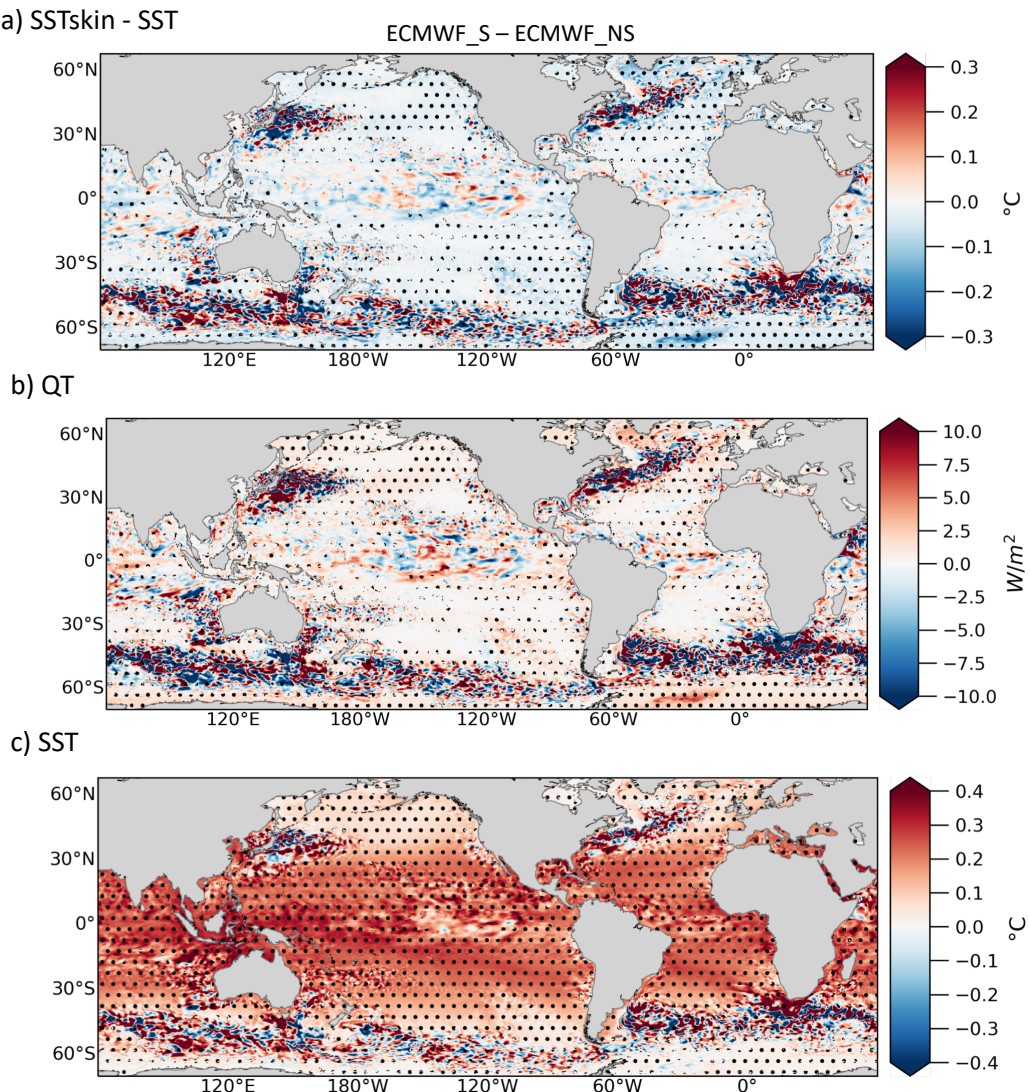

**Figure 5.** Annual mean differences of a) SSTskin-SST, b) total turbulent heat fluxes ($Q_T$) and c) SST between `ECMWF_S`- `ECMWF_NS`. Hatching indicates significant values (95% confidence level).

## 3.4 Drag coefficient and Wind stress

The impact of the wind stress in driving the SST differences between `ECMWF_S` and `NCAR` bulk parameterizations is here investigated by comparing results from `ECMWF_NS` and `CdNC_CeEC_NS` simulations (see Table 1 for experiments details). The SST simulated by `ECMWF_NS` is colder than `CdNC_CeEC_NS` over EBUS and the tropical Pacific and Atlantic oceans 270 (Figure 6b), regions characterized by wind driven upwelling. This suggests that wind stress is a major driver of the SST

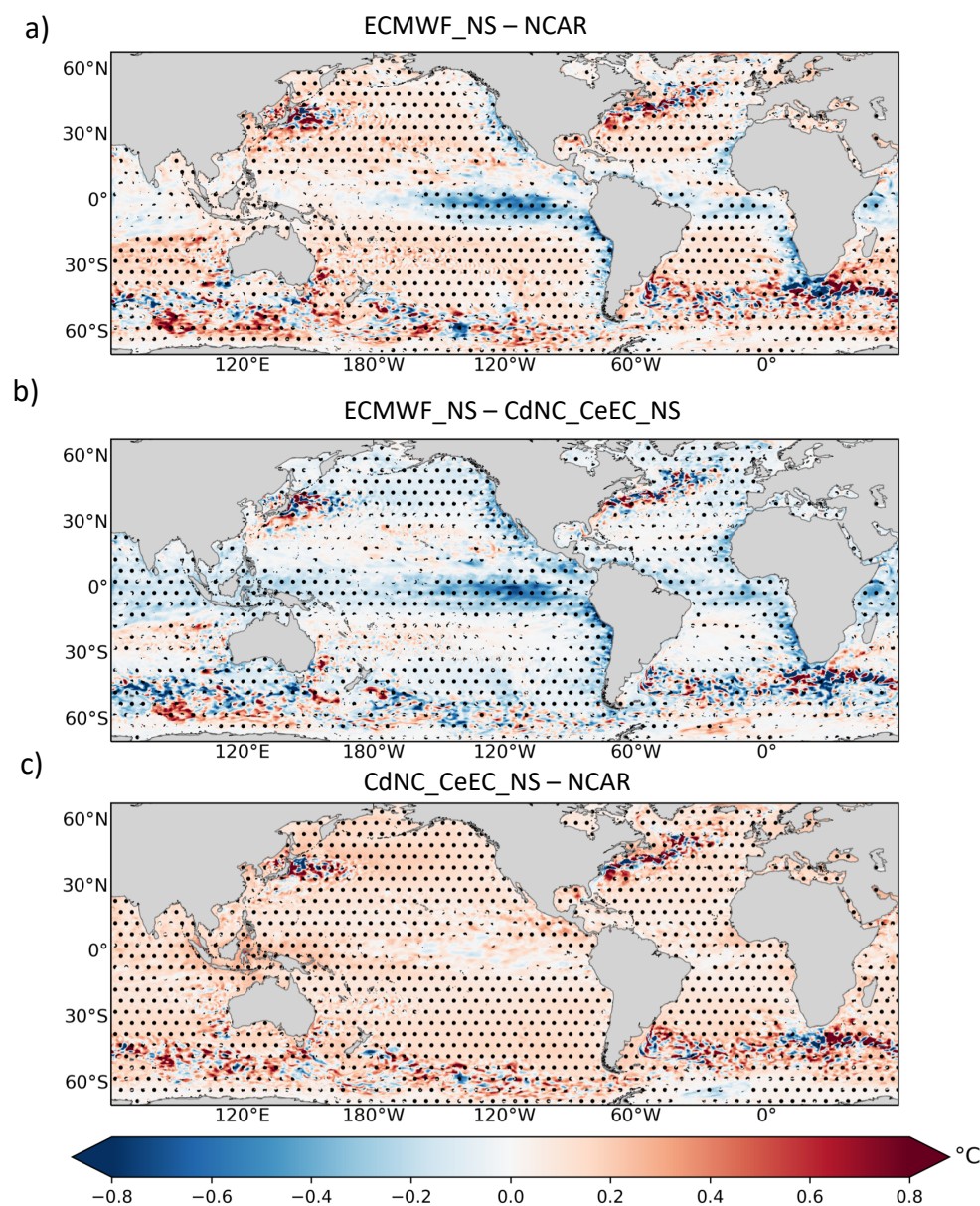

**Figure 6.** Annual mean SST differences between a) ECMWF_NS - NCAR, b) ECMWF_NS - CdNC_CeEC_NS, c) CdNC_CeEC_NS - NCAR. Hatching indicates significant values (95% confidence level).

differences (Figure 4a). Referring to Equation 1a, the wind stress is proportional to the wind speed vector at height z ($\mathbf{u_z}$), the bulk scalar wind speed $|\mathbf{u_z}|$ (with the potential inclusion of a gustiness contribution $U$), and the drag coefficient ($C_D$). Including gustiness in the ECMWF calculation produces the scalar wind differences in Figure 9a. As expected, the differences

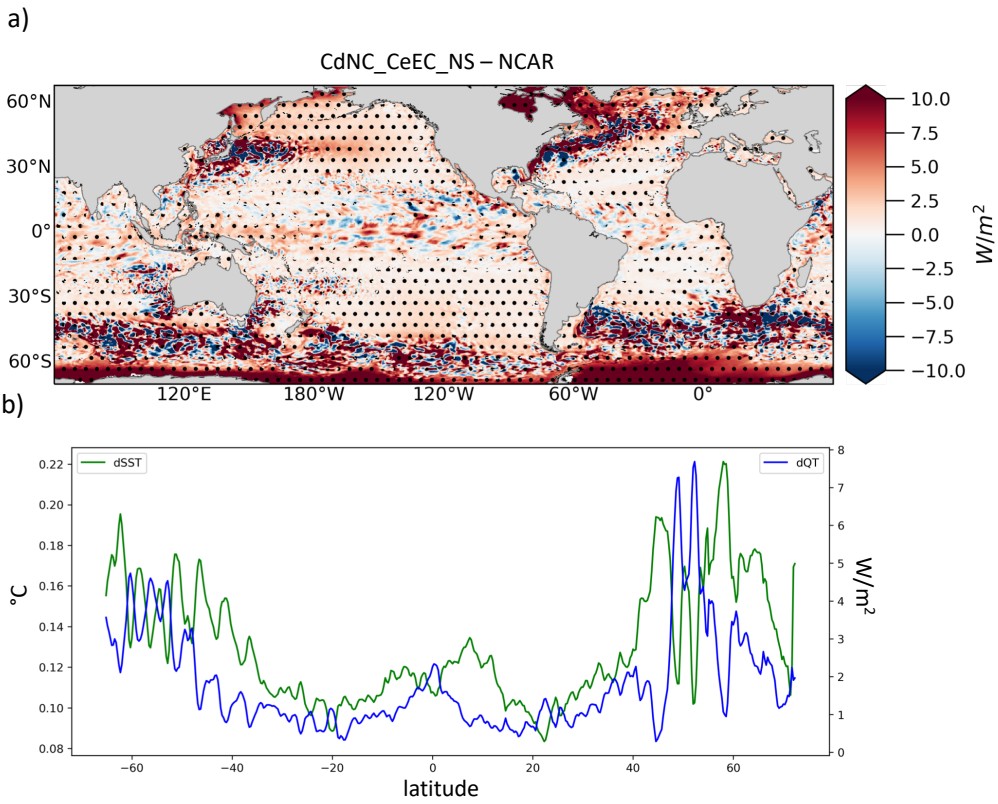

a)

CdNC_CeEC_NS − NCAR

b)

**Figure 7.** a) Annual mean differences of total turbulent heat fluxes $QT$ between `CdNC_CeEC_NS` and `NCAR` experiments. b) zonally-averaged differences of SST (green) and of $Q_T$ (blue) annual means between the same experiments. Hatching indicates significant values (95% confidence level).

caused by the gustiness correction emerge in regions with calm and unstable conditions. They are indeed located in the 5°N -

275    10°N latitude band, in the eastern Pacific and Atlantic oceans, in the tropical western Pacific including the southern China Sea and the tropical Indian Ocean. These differences do not exceed $0.3 m/s$.

Differences of $C_D$ and $C_D^{N10}$ fields between experiments show similar patterns (Figure 9b-c), suggesting that the differences in $C_D$ between parameterizations are related to the neutral coefficient ($C_D^{N10}$) calculation rather than to its stability correction (term to add to $C_D^{N10}$ to get $C_D$ coefficients). Indeed, as discussed in section 2.2 for $C_D^{N10}$, the ECMWF $C_D$ is larger than

280    NCAR for wind speeds above 5 m/s, smaller than NCAR for calm up to light breeze conditions ($U < 5$ m/s). In the areas where $U$ is approximately 4-5 m/s, such as in the north-west Pacific and Atlantic ocean (between 20°N and 30°N) and in the south-east Pacific and Atlantic ocean (between 20°S and 30°S), the ECMWF $C_D$ is similar or slightly smaller than NCAR.

Since the wind stress is not affected by the type of first-order feedback at play for the NSHFs (SST-$Q_T$ negative feedback, see section 2.2), differences of $U$ and $C_D$ between experiments are reflected onto the resulting different fields after bulk calculation

285    (i.e. $\tau$ and $WSC$, Figure 10). In particular, over the ACC, the northern and southern mid-latitudes (e.g. EBUS), and the Atlantic

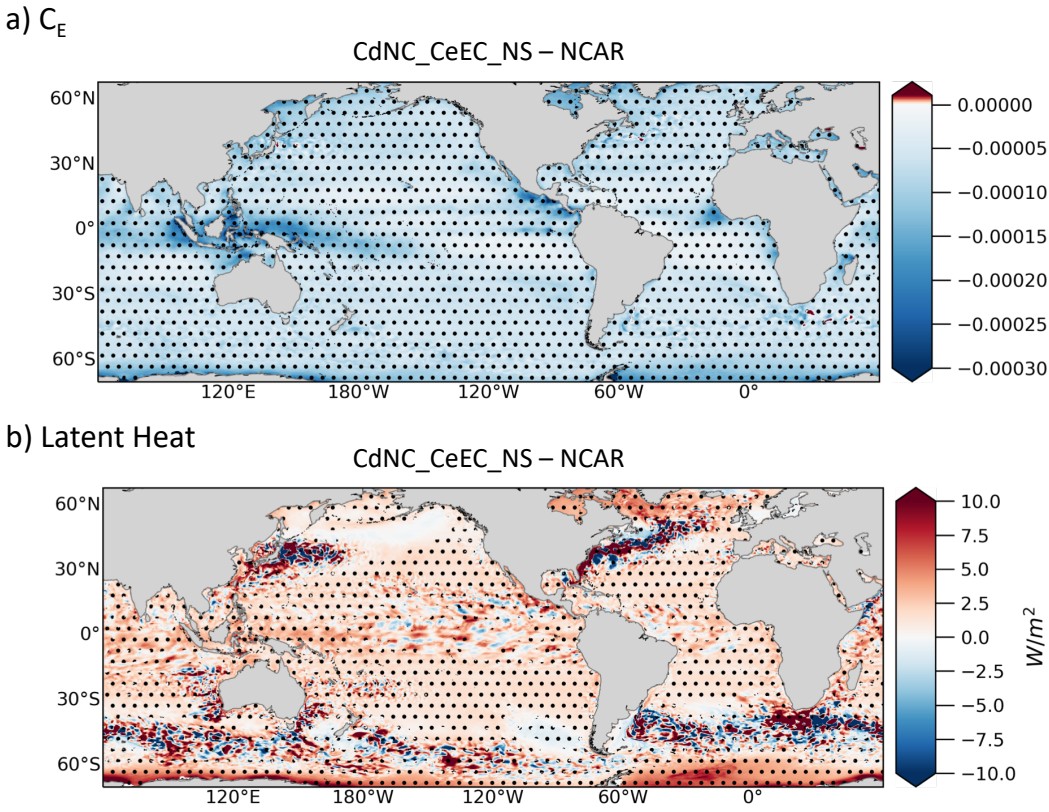

a) $C_E$

CdNC_CeEC_NS − NCAR

b) Latent Heat

CdNC_CeEC_NS − NCAR

**Figure 8.** Annual mean differences of a) specific humidity transfer coefficient ($C_E$), b) latent heat between CdNC_CeEC_NS and NCAR

storm track (i.e regions characterized by wind speeds above $5m/s$ and ECMWF_NS $C_D$ larger than CdNC_CeEC_NS $C_D$, see Figure 10), the ECMWF_NS wind stress is stronger by an average value of $0.035N/m^2$ (about 20% of NCAR absolute value) with respect to NCAR. In the 5°N - 10°N region, latitudinal band characterized by mean winds below $5m/s$ and $C_D$ larger in CdNC_CeEC_NS than ECMWF_NS (Figure 10), ECMWF_NS shows a wind stress reduction of -0.003$N/m^2$ (about 3% of NCAR absolute value) with respect to NCAR. In regions where the differences in $C_D$ and wind stress are opposite (e.g. the north-west and south-west Pacific and Atlantic ocean, Indian ocean, Baja California), the high time-variability of the $C_D$ differences (not shown) could hide the relation between $C_D$ and $\tau$. In addition, including the convective gustiness in $U$ calculation strengthens the wind stress in ECMWF_NS. Both hypotheses are verified, the ECMWF_NS experiment presents a stronger wind stress almost everywhere over the global ocean compared to a twin experiment (i.e. ECMWF_NS_NG) where the convective gustiness is not used in the computation (Figure S2) and the correlation between $C_D$ differences and wind stress differences is always significant and positive (not shown). The higher the difference in $C_D$, the stronger the differences in wind stress.

The SST differences between ECMWF_NS and CdNC_CeEC_NS over the tropical Pacific and Atlantic Oceans (Figure 6b) are likely related to Ekman suction, which is driven by the positive (negative) wind stress curl in the northern (southern) hemi-

sphere. Substantial differences are found in `ECMWF_NS` compared to `CdNC_CeEC_NS`, characterized by greater mean wind stress both north and south of the tropical band and weaker wind stress along the equator (Figure 10a). These latitudinal differences of the wind stress between experiments reflect in the differences in the wind stress curl patters (Figure 10b). Indeed, a stronger acceleration (deceleration) of southeast trades north (south) of the equator in `ECMWF_NS` may lead to a stronger positive (negative) curl north (south) of the Equator (Chelton et al., 2001). We found this relation significant north of the equator: the stronger positive wind stress curl in `ECMWF_NS` than `CdNC_CeEC_NS` results in a colder SST in `ECMWF_NS` compared to `CdNC_CeEC_NS` (see correlation map in Figure S3).

The stronger wind stress along EBUS in `ECMWF_NS` compared to `CdNC_CeEC_NS`, instead, likely enhances coastal upwelling, explaining most of the SST differences over these regions. Part of the SST difference could be also related to Ekman suction. `ECMWF_NS` shows stronger positive (negative) wind stress curl in the northern (southern) hemisphere EBUS compared to `CdNC_CeEC_NS` (Figure 10b). The vertical velocity, and in turn, the coastal SST along EBUS are, indeed, extremely sensitive to wind forcing changes (Bonino et al., 2019; Small et al., 2015; Capet et al., 2004; Desbiolles et al., 2014). These relations are confirmed along the coast of the Benguela Upwelling System (Figures S4 and S5). During the Benguela upwelling season (ONDJ), the enhanced wind stress and negative wind stress curl in `ECMWF_NS` reinforce the vertical velocity with respect to `CdNC_CeEC_NS` (Figure S4), resulting in colder surface temperature (see correlation maps Figure S5).

It is important to highlight that the differences in the wind stress are also responsible for the changes in the meridional heat transport. MHT differences between `ECMWF_NS` and `CdNC_CeEC_NS` resemble the differences between `ECMWF_S` and `NCAR` (compare Figure 4c and Figure 11c), with a higher transport in `ECMWF_NS` at all latitudes. The largest differences are located in the tropical region (up to 0.6 PW, about 18% of NCAR mean value), where the differences in meridional transport (linked to the equatorial upwelling) between the two experiments are likely maxima.

Even though the two experiments use the same $C_E$ and $C_H$, the dependence of $Q_L$ and $Q_H$ to the prognostic SST at each time-step generates differences in $Q_T$ (Figure 11a). The ocean gains heat in `ECMWF_NS` compared to `CdNC_CeEC_NS` (i.e. positive $Q_T$ differences) over the EBUS and the Equatorial region. In contrast to the previous finding, the differences in $Q_T$ and SST have opposite sign, indicating that SST differences drives the $Q_T$ differences: the colder the temperature produced by `ECMWF_NS` wind stress with respect to `CdNC_CeEC_NS`, the higher the heat gained by `ECMWF_NS` along the latitudes (Figure 11b). In summary, `ECMWF_NS` reproduces stronger wind stress and wind stress curl along EBUS, and stronger cyclonic wind stress curl along the Equator, that generates colder SST with respect to `CdNC_CeEC_NS`, through enhanced upwelling processes.

In light of the importance of the wind stress in driving the SST differences between ECMWF and NCAR parameterizations, we discuss why `COARE_S` does not display the cold SST differences in comparison to `NCAR` over EBUS and equatorial Pacific (Figure 4b). With wind speed ranging from 7 to 9 m/s (e.g. over EBUS) the $C_D$ in `COARE_S` parameterization is smaller than that of ECMWF parameterization, but slightly higher or almost identical (around 7 m/s) than the $C_D$ of NCAR (refer to Figure 1c). Moreover, over the northern equatorial band the $C_D$ of `COARE_S` is smaller than that of `ECMWF_S` and `NCAR`. As a consequence, the `COARE_S` differences in wind stress (Figure 3b) in comparison with `NCAR` are characterized by a strong decrease, roughly 10%, over the northern equatorial band and a slightly increase of the wind stress, roughly 2%,

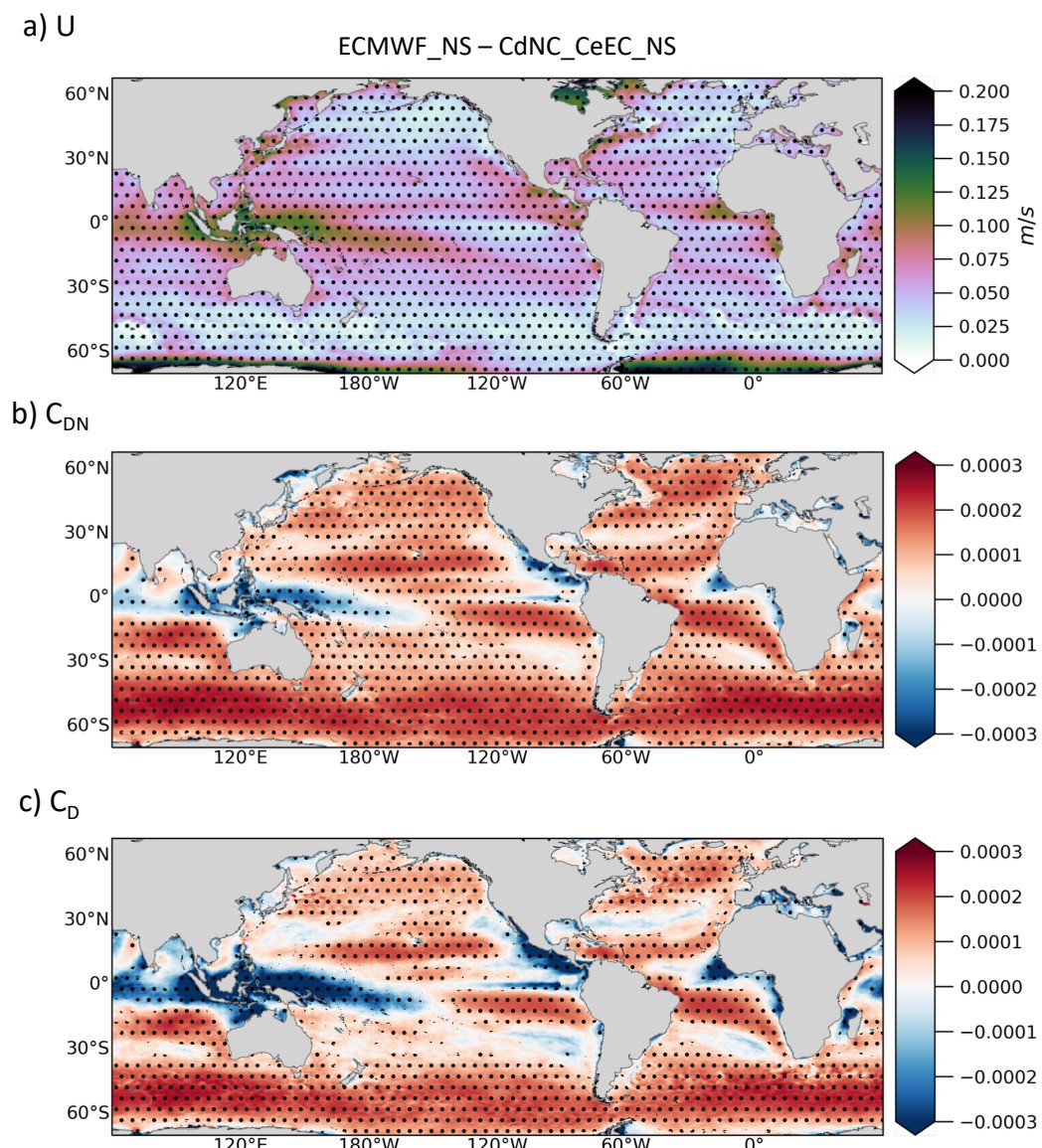

**Figure 9.** Annual mean differences of a) wind speed ($U$), b) neutral wind stress transfer coefficient ($C_{DN}$) and c) wind stress transfer coefficient ($C_D$) between `ECMWF_NS` and `CdNC_CeEC_NS`. Hatching indicates significant values (95% confidence level).

over EBUS. The increase of wind stress over EBUS in `COARE_S` (2% in comparison to 25% in `ECMWF_S`) is not enough to promote stronger coastal upwelling in the annual mean, and in turn colder SST with respect to `NCAR`. As regard the equatorial upwelling, the weak increasing of the wind stress in the north equatorial region (e.g. northern equatorial cold front, Figure 3b) compared to `NCAR` wind stress (Figure 3a), prevents the enhancement of the positive wind stress curl in `COARE_S` (Figure

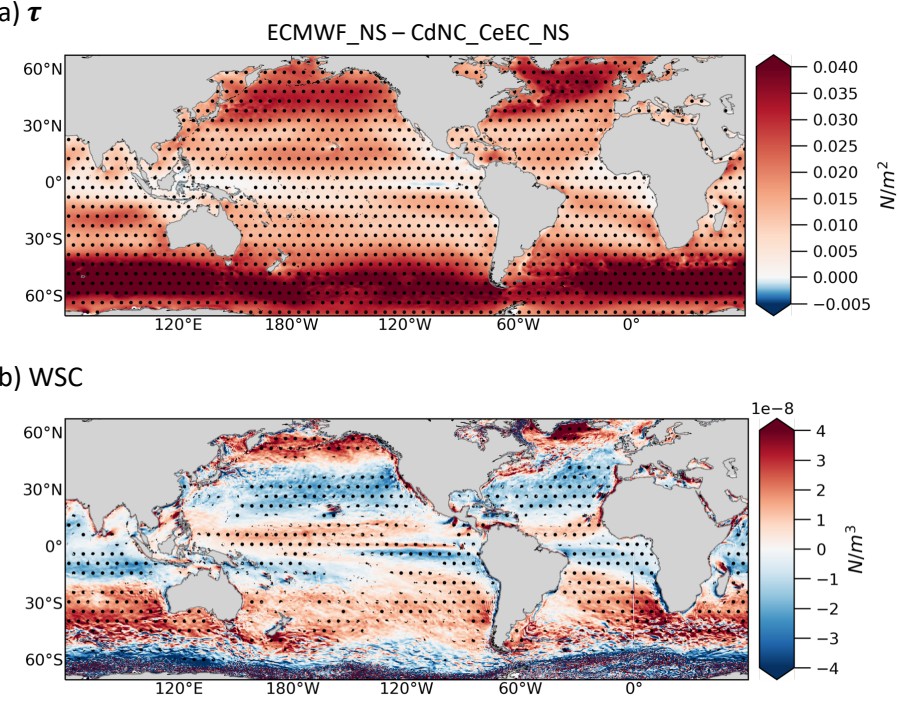

a) $\tau$

ECMWF_NS − CdNC_CeEC_NS

b) WSC

**Figure 10.** Annual mean differences of a) wind stress ($\tau$) and b) wind stress curl ($WSC$) between `ECMWF_NS` and `CdNC_CeEC_NS`.

3c). Nevertheless, to properly identify the drivers of the pattern in the SST differences between `COARE_S` and `NCAR` extra dedicated numerical experiments should be performed.

### 3.5 Online prognostic SST approach vs offline prescribed SST approach

In order to discuss the role of the SST-$Q_T$ negative feedback at play in the online prognostic SST approach, we compare our results with Brodeau et al. (2017), who compared the different bulk parameterizations using the offline prescribed SST approach (i.e. TASFs are computed by means of bulk formulas using prescribed surface atmospheric variables and prescribed SST). It is worth mentioning that there are few discrepancies in the bulk implementation between this study and Brodeau et al. (2017). They used the COARE3.0 parameterization instead of COARE3.6 and, their simulations, performed for a longer (1982-2014) period, are forced by the ERA-Interim reanalysis instead of ERA5. Therefore, our scope in this comparison is only to qualitatively understand the negative feedback between the SST and the $Q_T$ at play in our experiments.

Brodeau et al. (2017) report a mean global increase of the wind stress of $20mN/m^2$ using ECMWF parameterization instead of NCAR parameterization. The computation of the wind stress is not affected by the SST-$Q_T$ negative feedback (see equation 1a), so that our results of $20mN/m^2$ global mean increase of wind stress is completely in line with the prescribed SST comparison by Brodeau et al. (2017). Our findings do not follow Brodeau et al. (2017) in terms of the $Q_T$ differences between

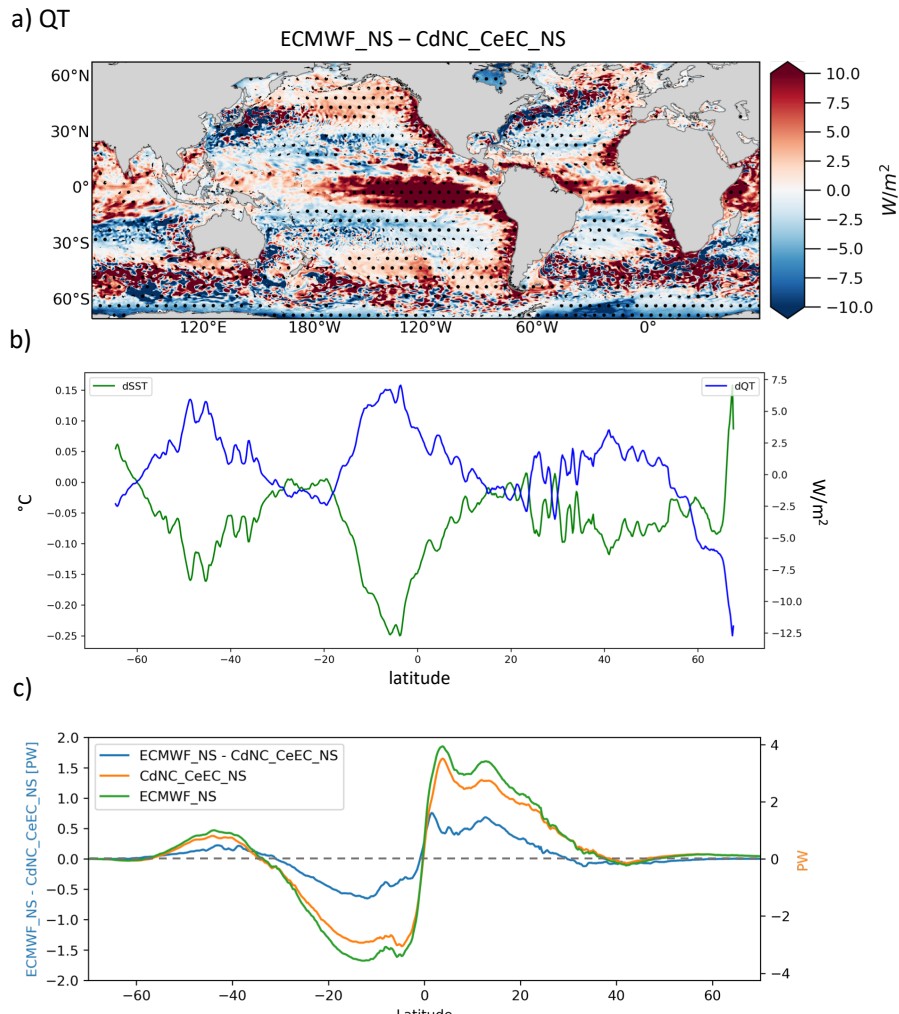

**Figure 11.** a) Annual mean differences of turbulent heat fluxes ($Q_T$) between ECMWF_NS - CdNC_CeEC_NS; b) Annual zonal-mean differences time-series of SST (green) and $Q_T$ (blue) between ECMWF_NS - CdNC_CeEC_NS; c) Global Meridional Heat Transport in the upper 100m ocean (values on the right y axis) for ECMWF_NS and CdNC_CeEC_NS and differences (values on the left y axis) between them. Hatching indicates significant values (95% confidence level).

ECMWF_S and NCAR parameterizations. They find a global mean increase of $Q_T$ of $13W/m^2$ for ECMWF_S, while in our experiments ECMWF_S displays a mean global increase of $5W/m^2$ with respect to NCAR. Moreover, they report an increase of $7W/m^2$ considering SSTskin rather than SST in COARE parameterization, while in our experiments ECMWF_S displays a mean global increase of $1W/m^2$ with respect to ECMWF_NS. The negative feedback between the SST and the $Q_T$ which is active in our experiments reduces the differences in the total turbulent flux across parameterizations compared to the prescribed SST comparison.

## 4 Summary and conclusions

In this work we have investigated how the implementation of different bulk parameterizations in NEMOv4 ocean general circulation model drives substantial changes in prognostic sea surface temperature. Specifically, we studied the contribution of different aspects and assumptions of the bulk parameterizations in driving the SST differences across numerical experiments performed using NEMO global model configuration with $1/4°$ of horizontal resolution. In particular, we analyzed and quantified the role of the inclusion of the skin temperature in the computation of the turbulent heat flux components, and we also studied the role of the turbulent heat flux components and of the wind stress in driving the SST changes between parameterizations. In order to do that, we analysed differences between 'control experiments', short-term numerical experiments which used the bulk parameterizations implemented in NEMOv4, and 'mixed experiments', short-term sensitivity experiments where bulk assumptions are excluded (e.g skin temperature) or bulk transfer coefficients are computed mixing the bulk parameterizations (e.g. $C_D$ from NCAR parameterization and $C_E$ and $C_H$ from ECMWF parameterization). Moreover, the relevance of this work, other than highlighting the sensitivity of the sea surface temperature to the bulk parameterizations, is also to discuss the role of the SST-$Q_T$ negative feedback at play in the simulations. As such, we compared the modeled turbulent air-sea fluxes with the estimations from Brodeau et al. (2017), who analyzed the same bulk parameterizations, but using offline prescribed SST approach. The findings can be summarized as follow:

1. The implementation of skin temperature in the bulk parameterizations reduces evaporation and decrease the turbulent heat flux to the atmosphere, promoting ocean warming. The skin temperature is usually colder than the sea surface temperature. The skin temperature contribution in terms of turbulent heat flux is weaker with respect to the Brodeau et al. (2017) estimations due to the negative feedback between the SST and the $Q_T$. In our experiments the SST is free to evolve and feeds back negatively with respect to $Q_T$.

2. The turbulent heat flux differences between experiments are dominated by the latent heat flux contribution, which arise from $C_E$ differences between bulk parameterizations. Less evaporative ocean gains heat, which tends to promote ocean warming. The turbulent heat flux differences are weaker with respect to the estimations of Brodeau et al. (2017) and they can be attributed to the SST-$Q_T$ negative feedback.

3. The wind stress differences between bulk parameterizations are attributable to the $C_D$ differences, which result crucial especially in wind-driven dominantly ocean regions. In particular, experiment with enhanced wind stress or wind stress curl over EBUS and over Equatorial Pacific promote upwelling processes and consequent cooling of the sea surface temperature. Stronger wind stress results in an increase of the poleward heat transport in the upper ocean, which a more pronounced increase in the $\pm20$ latitude band. The wind stress differences across the bulk parameterizations implemented in NEMOv4 is of the same magnitude of the wind stress differences calculated by Brodeau et al. (2017). This is due to the fact that, at first order, the wind stress computation is not affected by the SST.

It is worth underlining that we are using forced ocean experiments in which the atmospheric fields (e.g. wind, air temperature, air humidity) given to the ocean model and seen in the online prognostic SST approach come from an atmospheric reanalysis,

and do not respond back to the ocean variability. Introducing the air-sea feedback in the system might substantially impacts the turbulent fluxes and modify our finding in comparing the SST response among the bulk parameterizations. In the perspective of improving the representation of air-sea interaction in the NEMO framework, an atmospheric boundary layer (ABL) is integrated in the new NEMO release 4.2 (Lemarié et al., 2021). Nevertheless, the ABL implementation is in a preliminary stage and the current online prognostic SST approach is still the favorite. Please note that the new release of NEMO, v4.2, includes some modifications to the bulk formulas of the version used in this study. These changes do not affect the presented results.

## Appendix A: Table A1

**List of Acronyms and Symbols**

| Acronym | Expansion |
| --- | --- |
| TASFs | Turbulent Air-Sea Flux components |
| THFs | Turbulent Heat Flux components |
| NSHFs | Non Solar Heat Flux components |
| $Q_T$ | Total turbulent heat flux |
| BTC | Bulk Transfer Coefficient |
| SSTSkin | Sea Surface Skin Temperature |
| CSWL | Cool Skin and (diurnal) Warm Layer |
| EBUS | Eastern Boundary Upwelling Systems |
| MHT | Meridional Heat Transport |
| WSC | Wind Stress Curl |

*Code availability.* This version of the NEMO code is based on code release 4.0, revision number 12957 (https://forge.ipsl.jussieu.fr/nemo/browser/NEMO/trunk?rev=12957, last access: 24 February 2022). The original code was modified in the computations of the bulk transfer coefficients applied to perform the experiments. The code and the namelists to run each experiment are available in the Zenodo archive (ttps://doi.org/10.5281/zenodo.6258085, DOI: 10.5281/zenodo.6258085). The model outputs used to produce the figures are also available in the Zenodo archive.

*Author contributions.* GB modified the numerical code, set up the experiment and wrote the manuscript. DI conceived and designed this study. All authors contributed to the interpretation of results and editing.

*Competing interests.* The authors declare that they have no conflict of interest.

*Acknowledgements.* We acknowledge the CMCC Foundation for having provided computational resources.

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
