# Peer review of "The bulk parameterizations of turbulent air-sea fluxes in NEMO4: the origin of Sea Surface Temperature differences in a global model study"

_Geoscientific Model Development, 2021_

## Referee Comment (RC1)

Review of 'The bulk parameterization of turbulent air-sea fluxes in NEMO4: the origin of sea surface temperature differences in a global model study', by G. Bonino, D. Iovino, L. Brodeau and S. Masina, submitted to GMD.

**Summary**

The paper overviews the effect of using alternative air-sea flux parameterization schemes in the NEMO ocean-ice model. The schemes considered are NCAR CORE (large and Yeager 2009), ECMWF (2015) and COARE 3.6 (Fairall, Edson et al.). The focus is on the effect of using a surface skin temperature, rather than a bulk temperature, and on the effect of using different formulations for exchange coefficients (for momentum and heat). Model experiments with the different schemes are compared, as well as sensitivity experiments where aspects of the schemes are mixed. It is found that the schemes cause SST differences of O(0.1deg.C) , surface heat fluxes of O(1W/m2) (i.e. less than 10W/m2), wind stress changes of 10-20%, wind stress curl changes of maybe 10%. Reasons for the differences are explored using the sensitivity experiments. Finally, the results are compared to a previous work that performed similar experiments but using fixed (observed) SST and determined differences in heat flux and momentum flux.

The paper is clear, the experiments well-designed, and the result sections are mostly well-written. The sensitivities are quantified, and the comparison with the fixed SST experiments is interesting. The results are important to know, for climate and/or ocean applications. However I suggest major revisions before publication based on the major comments below and have further minor comments

**Major Comments**

The paper shows differences of certain fields (SST, heat fluxes, momentum fluxes etc.) between the experiments. Is it possible to say whether any of the cases are more realistic than other , compared to observations, or is it complicated by competing and possibly cancelling effects of other parameterizations or processes? Could you look at other ocean variables (like the surface flow) to help with this?

Can the results be put in context by comparing with known sensitivities to other well-known parameterizations or processes? Do the results have any impact on meridional heat transport?

**Minor Comments**

Line 5 – "calculate"

Delete (revision 12957), move this to Methods.

Line 16. "weaker than that found by"

Line 20. Isn't surface radiative flux also highly important???

Line 78. Re "Marsaleix et al." – based on the title of this paper, it does not obviously mention TKE, but it does mention energetics. Please confirm it is the correct reference. Sorry, I have not read it.

Line 98 "possibly referred to the ocean currents" -> "which may be referred to the ocean currents"

At this point, the reviewer might anticipate experiments to look at the effect of including surface currents in stress. Your paper does not do this, which is fine, but you may want to refer to the extensive literature on the subject (e.g. Renault et al., Sun et al, and many others).

Line 100. "increase"

List starting Line 110. I would add:

3. Effect of including ocean current in stress

4. The form of the exchange coefficients

Then you can mention which of these effects you look at. Am I correct in thinking you do not explicitly look at the effect of convective gustiness? See comment later.

Line 115-116. "is indeed influenced by the" -> "uses the"

Line 119 I suggest "QT increases in response, then the SST will tend…"

Line 128 . NCAR scheme … minimum wind speed of 0.5 m/s …  This is interesting, and I just confirmed this is also done in the CESM scheme. Note that the Large and Yeager drag coefficient actually goes to infinity as you approach zero wind speed (your Fig. 1b). So even if the wind speed gets very low, the momentum flux remains significant.

Lines 129-134. I would say that the Large and Yeager scheme also uses MOST. It combines MOST theory with a semi-empirical form of drag coefficient.

Fig. 1b. I understand that you do not focus on high/extreme wind speeds, but I am curious to know what happens above 35m/s. There is some discussion on this topic in Fu et al. (2021), their sections 3.2 and 4.1. (Note that their paper employs the original Large and Yeager (2004) form of drag coefficient, without reduction at high wind speeds.) Note also that ERA5 is a high-resolution dataset and will include extreme wind events. Reference: Fu, Dan et al. 2021: Introducing the new Regional Community Earth System Model, R-CESM. B. Amer. Meteor. Soc., 102, E1821-E1843, https://doi.org/10.1175/BAMS-D-20-0024.1

Lines 134-145. It would be useful to show a zoomed-in plot of Fig .1b for winds 10m/s or less. Also, replace "promote" with another word like enhance or increase.

Line 182. Fig. 2 label shows total surface heat flux, should be turbulent surface heat flux, if I understand the terminology.

Line 221. Fig. 6a shows very small QT (~1 W/m2) over most of the Globe, only small regions reach 10W/m2.

Line 224. Replace with "parameterization, is smaller than the CE of NCAR…" "This leads to an underestimation of evaporation, increased input of heat,…"

Line 236. Fig. 5a->Fig. 5b ?

Line 258-259. I believe you do not explicitly look at the sensitivity to convective gustiness parameterization.  So your inferences here are solely based on the fact that CD differences are small in these regions? You can consider running an extra sensitivity experiment with convective gustiness switched off in ECMWF.

Lines 259 onwards. It is not obvious to me from Fig. 8 that the EBUS will be notable regions of enhanced stress and WSC. Perhaps zoom in on an example EBUS and show the causal links more clearly between U, CD, TAU and WSC.

On this topic, the lead author has 2 nice papers on EBUS in JRA55do and ERA-Interim-forced runs. Based on this experience, can you comment on whether the changes to TAU and WSC are realistic and whether they would make a sizeable change to upwelling?

Line 268. Can you see any changes to the North Equatorial Undercurrent, which is mainly an WSC-driven system (e.g. Sun et al. 2021 and references therein, https://doi.org/10.1016/j.ocemod.2021.101876 )

Line 286. But the equator to 10deg. N difference is still large with COARE (Fig. 2a, right)

Line 308. "Namely"-> "In particular"

Line 338 "prospective"-> "perspective"

Line 341-3342. Delete sentence starting "Currently", it can be understood from lines above.

---

## Referee Comment (RC2)

**Comments to gmd-2021-436**
**"The bulk parameterizations of turbulent air-sea fluxes in NEMO4: the origin of Sea Surface Temperature differences in a global model study" by Bonino et al.**

The authors exploit a state-of-the-art ocean model to evaluate the differences in three algorithms of bulk air-sea turbulent fluxes. The use of an ocean model enables to have a dynamical SST that modulates the fluxes, so that the air-sea coupling is partially represented. The atmospheric response to this SST forcing is, in fact, still missing.

The work is surely of interest, timely and well structured. Apart from some English editing and some improvements of the figures, to increase their readability, there are few major points (see the 'Major comments' section) that should be addressed before being accepted. In particular, some more analyses of the oceanic response to the different forcing should be included, to fully take advantage of the information provided by these heavy simulations. For these reasons, I suggest major revisions.

**Major comments**

A lot of effort is put in explaining quite successfully why there is a cold bias in the equatorial eastern Pacific and the EBUS regions between ECMWF and NCAR bulk schemes. However, very little attention is given to the strong differences in heat fluxes found over western boundary current (WBC) systems, which are known to be areas with strong air-sea interactions (See figure 2 of the manuscript). I invite the authors to dig a bit more in this direction and explore if there is a meaning in the spatial and temporal variability of this difference. If it was only noise, I would expect it to average to zero in an annual mean, but, indeed, it is visible in both ECMWF-NCAR and COARE-NCAR differences (figure 3).

What are the limitations in running simulations that last one year only? Is there any dependence on the specific year (e.g. in terms of ENSO phase, or any other climatic mode)? What about the spinup of the model? Which year has been considered?

With respect to Brodeau et al. (2017, B17 hereafter), the fluxes are computed using a dynamical SST field that responds to the atmospheric forcing. However, the atmospheric dynamics is known to respond to the SST even on daily and sub-daily time scales (see the review of Small et al., 2008, and some examples of applications in different areas of the world such as Li and Carbone, 2012; Gaube et al., 2019; Desbiolles et al., 2021). It would be interesting to discuss a fully coupled approach, as it has been shown that surface winds and clouds are affected by the SST structure on daily time-scales which, then, affect the SST and the surface turbulent fluxes back. This is only mentioned at the end of the manuscript and it should probably be included in the Introduction, as well. Moreover, the closed loop of this kind of ocean-atmosphere interactions has been proposed to be responsible for a three-to-six day oscillation (Strobach et

al., 2020): I wonder if these oscillations are also observed here and whether they depend on the flux parameterizations.

In general, the fact that full ocean simulations are performed seems a bit underexploited. I think that much more information could be extracted, for example when discussing the role of different wind stress and wind stress curl in controlling the surface cooling in the EBUS and equatorial regions. Would it be possible to disentangle the role of upwelling and the role of entrainment in this surface cooling? What about doing some heat budget in the oceanic mixed layer to understand what processes are mostly modified by the different bulk algorithms?

The statistical significance of the differences between the experiments should be assessed. If the distributions are Gaussian, a t-test should be enough.

There are various differences with respect to the estimates shown in B17. In particular:
1. the authors consider a single year, whereas B17 consider the period 1982-2014;
2. the authors use ERA5 data to force NEMO, whereas B17 use ERA-Interim data;
3. COARE 3.5 is used here, and COARE 3.0 is used in B17;
4. Different versions of the ECMWF model are considered (cycle 40 and 41).

For these reasons, I would be more cautious in comparing the present results with those presented in B17. Would it be possible, for example, to compute the heat fluxes using the local midnight SST throughout the day, to mimic the fixed-SST approach, as in B17, and compare these fluxes to the prognostic-SST ones? This would avoid all the limitations highlighted before, as the original data would be the same.

**Minor comments**

There are typos throughout the text: I suggest a careful reading of the manuscript.

Many maps are hard to read, because the contour lines often mask the color shading. I suggest:
- enlarging the maps (as currently done in Fig 5, at least);
- reducing the number of contour lines (or removing them, if not necessary);
- verifying that the contours are properly plotted and not broken at 180° or 0° longitude;
- removing the word 'exp =' in the titles, as it is redundant.

**Technical comments**

The authors use both the words 'parameterization' and 'parametrization'. I suggest choosing the former and keeping it consistent throughout the text. Note that also the verb 'parameterise' is used in some places.

Abstract: specify the NEMO acronym.

L28 (and elsewhere): I would avoid the word 'state', as it has a specific thermodynamic connotation. I would simply refer to wind, air temperature and humidity as 'surface atmospheric variables'.

L32: One might want to add the reference to the latest version of the COARE 3.5 algorithm, namely Edson et al. (2013).

L50: It is true that with the prognostic SST approach, there is a negative feedback between the heat fluxes and the SST, but having a dynamical ocean can also modify the heat fluxes in the other direction. Namely, the heat fluxes can be strengthened (in absolute value) with the upper ocean mixing. Is there a way to disentangle these two contributions?

L76: 'Laplacian' should be capitalized.

L77: 's-1' should be 's$^{-1}$'.

L84 (and L148): The reference to ERA5 should be updated to Hersbach et al. (2020).

Eq (1a) should contain $U$ instead of $u$.

L97: $Q_T$ is dominated by $Q_L$ because $Q_L$ is much larger than $Q_S$. I wonder if the buoyancy flux, in which the sensible and the latent heat flux terms are comparable, is a more appropriate variable to consider. A recent example of its dynamical importance is the work by De Szoeke et al. (2021), where the buoyancy flux is shown to control the low-level cloud formation in the tropical Indian Ocean. This, then, has a significant influence on the surface fluxes.

L100: Replace 'increased' with 'increase'.

L103: An article 'the' has been misspelled.

LL106-107: The explanation of the cool-skin effect is not very clear.

L115: As the atmospheric component is not dynamically coupled to the ocean model I would modify this sentence: the air-sea feedback loop is partially closed.

L120: Is the word 'divergence' used to indicate the difference between the model formulations? If so, please replace it because it has specific meaning in vectorial calculus, which is a bit misleading.

L124: Edson et al. (2013) introduced COARE 3.5 and not 3.6. This should be corrected throughout the manuscript.

Fig1: Instead of using thin lines for the moisture transfer coefficients, thick dashed lines would be more visible and easier to distinguish from the drag coefficients. I would also suggest

reducing the range of wind speed in panel (b) up to 22 or 25 m/s, as the focus is on the left side of the panel.

L134: Repetition of the word 'of'.

LL134-144: There are several typos or missing articles and commas in this paragraph, please revise.

L148: Remove 'Weather'.

L151: The word 'uses' is chopped.

LL150-163: I would suggest removing the bullet points and use plain text, instead, to remove the repetitions and enable a smoother reading. Table 1 is already giving a schematic recap of the experiment setup. It is also not clear what is the difference between the parameterizations that use the absolute wind speed (as in experiments ECMWF_S, COARE_S, ECMWF_NS and CdNCAR_CeEC) and the parameterization that does not include the current correction (as in the NCAR experiment). It seems that the ocean surface currents are never used in this set of experiments (L169). Thus, it can simply be stated once, as this is not a parameter that changes. I also find the names of the experiments very confusing: what about making them more explicit with something like: CdEC_CeEC (instead of ECMWF_S), CdCO_CeCO (COARE_S), CdNC_CeNC_NS (NCAR), CdEC_CeEC_NS (ECMWF_NS), CdNC_CeEC_NS (CdNCAR_CeEC)? In this way, the differences among them are readily available in their names.

Table 1: What about adding 'Experiment name' in the first row of the first column? A column indicating whether the gustiness in the computation of the wind stress is included would be useful.

Figure 2 is not described in the main text, please do. What is its link with the SST bias? The name of the experiments should be kept consistent throughout the text. Here, for example, 'COARE3.6_S' should be replaced with 'COARE_S', or its correct name. It should be clarified (and motivated) whether the annual mean of the percentage difference or the percentage difference of the annual means is computed.

LL185-192: This paragraph is rather general and could be moved backward in the manuscript. One would expect here to find a reasoning on figures 2 and 3, such as why such patterns are observed, which specific reasons could explain them and their relationship, etc.

Fig4: The skin SST effect has a component at the daily scale. I wonder if, by considering the annual mean, the signal averages to zero. What about computing the temporal standard deviation of the difference SSTskin-SST?

LL202-207: The link between the figure and the text is not fully clear. By looking at the figure one might think that, on the annual average, there is an increase of SST when using the

SSTskin correction (is the sign of the difference ECMWF_S-ECMWF_NS correct?), because of a dominant diurnal warming effect. This is in contrast with the statement that the cool skin effect is dominant over the warm layer one. Is the annual mean computed using hourly outputs? How is the mean warming interpreted? What about its spatial structure?

L216: Are you referring here to the ECMWF_S or the ECMWF_NS experiment? Because, from the titles of the figures, the experiment ECMWF_S is considered in figure 3a, whereas the experiment ECMWF_NS is considered in figure 5a. Please clarify.

L218: The sentence is not grammatically correct and there is a repetition of the word 'differences'.

LL229-230: Is it 'higher heat absorption' or 'weaker heat loss'? The logical link between the latent heat considerations and the fact that it is the wind stress to be responsible for the observed cold SST pattern difference between ECMWF_S and NCAR is not clear.

L233: 'Computing'?

Fig6: Panel b) is not showing time series: the caption of the figure should be modified. From this figure one would not say that the mean excess QT is 10W/m2, as stated at line 221: where does this amount come from?

L239: Up to now, it is not very clear which parameterizations use the gustiness correction in the computation of the wind stress. As noted above, this information could be included in Table 1 and some more details on how the gustiness is included in the scheme should be given.

L240: 'Caused by the gustiness correction'.

Fig8: I suspect that the gustiness correction is highly variable in time on daily or even sub-daily scale. As for the CSWL correction, thus, I am not sure that showing the annual mean of such variables is enough. Wouldn't it be of interest to show the variance or the RMSE of the two model setup to better display where this highly variable correction is relevant?

L254: There seems to be an extra '(' at the end of the line.

L260: As the outputs of the ocean model are available, would it be possible to quantify the contribution of the surface cooling due to the modified Ekman suction between the configurations? Can a scaling between the anomalous wind curl and the anomalous SST cooling be derived?

LL277-289: By looking at figure 2 one would expect stronger differences in the SST in the COARE-NCAR comparison, and not in the ECMWF-NCAR one. What about showing the mean difference of the wind stress (not in percentage) and, maybe, the mean difference in wind stress curl, as it relates to the upwelling? Then, is there a contribution to the surface cooling from an

increased entrainment of cold waters in the OML (oceanic mixed layer) because of a stronger wind stress?

Fig9: Panel (a) is it the annual mean of the percentage variation or the percentage variation of the annual mean?

L342: 'favorite'.

**Additional references**

- Desbiolles et al. (2021) Links between sea surface temperature structures, clouds and rainfall: Case study of the Mediterranean Sea, *Geophysical Research Letters,* https://doi.org/10.1029/2020GL091839
- De Szoeke et al. (2021) Diurnal ocean surface warming drives convective turbulence and clouds in the atmosphere, *Geophysical Research Letters*, https://doi.org/10.1029/2020GL091299
- Gaube et al. (2019) Satellite observations of SST-induced wind speed perturbation at the oceanic mesoscale, *Geophysical Research Letters*, https://doi.org/10.1029/2018GL080807
- Li and Carbone (2012) Excitation of rainfall over the tropical western Pacific, *Journal of the Atmospheric Sciences*, https://doi.org/10.1175/JAS-D-11-0245.1
- Hersbach et al. (2020) The ERA5 global reanalysis, *Quarterly Journal of the Royal Meteorological Society*, https://doi.org/10.1002/qj.3803
- Small et al. (2008) Air-sea interactions over ocean fronts and eddies, *Dynamics of Atmospheres and Oceans*, https://doi.org/10.1016/j.dynatmoce.2008.01.001
- Strobach et al. (2020) Three-to-six-day air-sea oscillation in models and observations, *Geophysical Research Letters,* https://doi.org/10.1029/2019GL085837

---

## Referee Comment (RC3)

**Comments on " Bonino, G., Iovino, D., Brodeau, L., and Masina, S.: The bulk parameterizations of turbulent air-sea fluxes in NEMO4: the origin of Sea Surface Temperature differences in a global model study, Geosci. Model Dev. Discuss. "**

This study focuses on the SST sensitivity to surface turbulent fluxes computation differences using various bulk parameterizations and by including or not specific processes (skin temperature and wind gustiness). The SST sensitivity is assessed using global 1/4° forced ocean simulations, which is comparable to climate models resolution.

This work is of great interest for the ocean modelling community, but also for the climate modelling community, because of the big uncertainties associated with surface turbulent fluxes estimates, and their role on ocean-atmosphere energy transfer. A original and interesting aspect of this work also relies on the quantification of the negative SST-STHF fluxes. Despite these positive points, I suggest hereafter some major modifications to this study in order to improve the manuscript.

**Major comments**

A first important caveat of this study is the duration of the simulations, and consequently the significance of the results presented here. Even if the SST adjusts quickly and locally to surface turbulent fluxes modifications (from a few hours to a few days), the large-scale patterns and differences presented here might need more than one year to spin up and reach an new equilibrium state. The simulated interannual variability can also be modified between the sensitivity experiments, which can be misleading when interpretating the simulation differences. Another less important consequence is that spatial figures are quite noisy, which make them less readable.

Hence I would suggest to extend the different simulations to at least a 5-year period to make the results presented here more robust. As a comparison, Brodeau et al 2017 simulations which are referred in this manuscript discussions cover a 30-year period. If it not possible to extend the simulations for practical/technical reasons, I recommend to extend at least one simulation and compare the simulated turbulent fluxes and SST between the 1-year and the 5-year simulations to make this study more convincing.

Another important issue is that the manuscript does not contain any validation of the simulated oceanic state, and especially the SST. I understand that a detailed validation is out of the scope of this study, but SST is the only assessed oceanic variable here, and because it is a key variable in STHF computation, we should know about the potential model biases compared to observations, and how these SST biases can modify STHF estimates (through air-sea temperature and humidity differences), and more importantly the turbulent fluxes sensitivity. SST is a well observed variable, especially at global scale and over large time period, so it would not require too much work to include an observed SST map over the same period as a reference. The idea here is not to classify the "best" bulk parameterizations, but to have a global idea of model SST biases.

My last major point concerns the surface current effect into the surface stress computation. Considering absolute or relative winds in stress formula in forced ocean simulation is still debated, but an additional sensitivity experiment using relative wind could give additional insight (as it is done for wind gustiness for example) to this manuscript compared to Brodeau et al. 2017. From my understanding, as the prognostic SST does not influence surface stress (or very weakly through stability functions), your sensitivity experiments using different Cd is totally similar to Brodeau et al. 2017, and hence leads to the same already-known results. This additional experiment would allow to assess the current-stress negative feedback (as it is done for SST-STHF feedback in 3.5), and how it changes the stress sensitivity to the bulk choice. This would substantially enrich the 3.4 section of the manuscript, which is currently of limited interest.

**Minor comments**

A lot of English typos can be found in the text. A careful check is needed. Some punctuations are also missing.
Spatial figures must be improved to reach publication quality requirements. Here is some recommendations to improve them:
Spatial figures color extremes are often too much saturated and iso-contours are too thick. They are also very noisy due to the short experiments length. All these aspects make them hardly readable. Latitudes should also be extended from 70°S to 70°N as in Brodeau et al. to facilitate the comparison between those 2 studies.
A longitudinal average would also greatly improve and simplify figures interpretation as most of the results are mainly latitude dependant.
Some figures have resolution issue and appears blurry when zoomed in.

---

## Author Comment (AC3)

**REFEREE #1: JUSTIN SMALL**

Dear Dr. Justin Small,

we would like to thank you for the careful reading of the manuscript and the constructive comments that substantially helped to improve and clarify the paper. Answers to all your comments are detailed hereafter. Corrections to the English grammar were adopted in the revised manuscript according to the reviewer's recommendations, but they are not reported or discussed here. All authors agree with the modifications made to the manuscript. The comments by the referee are reported in bold followed by our responses (in blue). The text added to the revised manuscript is reported in italic font. The revised manuscript that includes track changes and line numbers is provided in pdf format.

Please note that, in this document, we use 'Figure' to identify the figures in the updated manuscript, while we use 'Plot' to identify the figures in this document.

The name of the experiments have been improved (following suggestions by the referees) as reported in Table 1. New names are used in this document and in the updated manuscript.

| Experiment name | sea surface temperature used ($T_s$) | computation of $C_D$ | computation of $C_E$ and $C_H$ | convective gustiness |
|---|---|---|---|---|
| COARE_S | SSTskin | COARE3.6 | COARE3.6 | Yes |
| ECMWF_S | SSTskin | ECMWF | ECMWF | Yes |
| NCAR | SST | NCAR | NCAR | No |
| ECMWF_NS | SST | ECMWF | ECMWF | Yes |
| CdNC_CeEC_NS | SST | NCAR | ECMWF | Yes |
| ECMWF_NS_NG | SST | ECMWF | ECMWF | No |

**Table 1.** Summary of the numerical experiments.

**Major Comments**

**1) The paper shows differences of certain fields (SST, heat fluxes, momentum fluxes etc.) between the experiments. Is it possible to say whether any of the cases are more realistic than other, compared to observations, or is it complicated by competing and possibly cancelling effects of other parameterizations or processes? Could you look at other ocean variables (like the surface flow) to help with this?**

We agree with the referee that a detailed comparison against observation can benefit the manuscript. Focusing on the sea surface temperature (SST), we compared the annual mean SST from the "control experiments" against the European Space Agency (ESA) Climate Change Initiative (CCI) SST (Merchant et al. 2019). Results are shown in the following Plot 1 (included as Figure 2 in Section 3.1 of the revised manuscript). Text has been added from lines 193 to 201: "*We compare the SST simulated by the*

*ECMWF_S, COARE_S and NCAR control experiments with the European Space Agency (ESA) Climate Change Initiative (CCI) SST dataset v2.0 (hereinafter ESA CCI SST dataset) which consists of daily-averaged global maps of SST on a 0.05° x 0.05° regular grid, covering the period from September 1981 to December 2016 (Merchant et al., 2019). All the control experiments present a warm bias in the Eastern Pacific, in the Eastern Boundary Upwelling systems (EBUS), in the Western Boundary Currents (WBCs) and in the Antarctic Circumpolar Current (ACC) region. The SST reproduced by COARE_S and ECMWF_S shows a cold bias of about -1°C in the North Atlantic open ocean at mid- latitudes, and a warm bias of about 0.5°C in the Indian Ocean and the Western Pacific (Figure 2a,b); NCAR SST is also colder than observations, with a larger bias of about -2°C in the North Atlantic (Figure 2c). The bias is generally higher compared with other two experiments and covers wider areas. "*

[Figure]

Plot 1: Annual mean SST differences between a) ECMWF_S, b) COARE_S, and c)NCAR against ESA CCI SST.

**2) Can the results be put in context by comparing with known sensitivities to other well-known parameterizations or processes? Do the results have any impact on meridional heat transport?**

The reviewer's suggestion is really interesting, but we think it goes somehow beyond the scope of the present work. The impact of bulk parameterizations on air-sea processes has been put in a more general context in a new paragraph added in the introduction (lines 39-46): " *It is worth mentioning that the online prognostic approach does only partially close the air-sea feedback. Surface winds and clouds are affected by the SST structure on daily time-scale which, in turn, affect the SST and the TASFs (Desbiolles et al., 2021; de Szoeke et al., 2021; Gaube et al., 2019; Li and Carbone, 2012; Small et al., 2008). The closed air-sea feedback (hereinafter coupled approach) in the system might substantially impact the turbulent fluxes (Lemarié et al., 2021; Small et al., 2008), but the coupled approach is still not yet mature in the ocean model community. Recently Lemarié et al. (2021) implemented a first attempt of a simplified atmospheric boundary layer model (ABL) to improve the representation of air-sea interactions in NEMOv4.2. However, the online prognostic SST approach is still largely used by the ocean modeling community in a variety of applications.*"

To quantify the impact of modified formulations on the meridional heat transport (MHT) , we computed it in the upper 100m of the global ocean and analyzed the differences among experiments (in Plot 2). The MHT in the ECMWF experiments is always higher compared to experiments that employ $C_D$ from NCAR formulation. ECMWF (with and without skin temperature) and NCAR-based experiments present the largest differences (Plot 2 a,e) that peak generally in the tropical band (about 0.8 PW, 20% of NCAR absolute value) where ECMWF wind stress is stronger than NCAR one. Global MHT in COARE_S and NCAR runs is comparable (Plot 2b), with differences lower than 0.3 PW. The transport differs only in the tropical latitude band in all the experiments that used the same $C_D$ (i.e. ECMWF_S and ECMWF_NS; NCAR and CdNC_CeEC_NS), and they are quite small (about 0.1 PW, Plot 2 c ,d).

Plot 2 (a,b) has been included in Section 3.3 of the revised manuscript, as Figure 4c,d; Plot 2e is in Section 3.4 as Figure 11c.

The following text was added to describe Figure 4 (lines 219-225): "*Changes on the simulated SST can reflect on the temperature profile in the upper ocean and the distribution of heat on global scales. We have computed the global ocean heat transport in the upper 100 meters and compared it among experiments. Figure 4 (c,d) presents the meridional heat transport (MHT) as a function of latitude. The MHT is larger in ECMWF_S compared to NCAR mostly at all latitudes (Figure 4c), with the largest differences (about 0.8 PW, 20% of NCAR absolute value) in the tropical band where ECMWF wind stress is stronger than NCAR one (Figure 3a). COARE_S and NCAR compare well, with differences lower than 0.3 PW (Figure 4d). Then, we will focus only on the differences between ECMWF_S and NCAR to analyze in detail the relationship between TASFs and SST. We show differences in MHT only when relevant.*"

The following text added in Section 3.4 (lines 313-317) describes Figure 11c: " *It is important to highlight that the differences in the wind stress are also responsible for the changes in the meridional heat transport. MHT differences between ECMWF_NS and CdNC_CeEC_NS resemble the differences between ECMWF_S and NCAR (compare Figure 4c and Figure 11c), with a higher transport in ECMWF_NS at all latitudes. The largest differences are located in the tropical region (up to 0.6 PW, about 18% of NCAR mean value), where the differences in meridional transport (linked to the equatorial upwelling) between the two experiments are likely maxima*"

In addition, we included the following text in the conclusion section (lines 384-385): *"Stronger wind stress results in an increase of the poleward heat transport in the upper ocean, which a more pronounced increase in the ±20 latitude band."*

[Figure]

Plot 2: Global Meridional heat transport in the upper 100m ocean (values on the right y axis) and differences (values on the left y axis) between a) ECMF_S and NCAR, b) COARE_S and NCAR, c)ECMWF_S and ECMWF_NS, d) CdNC_CdEC_NS and NCAR, and e) ECMWF_NS and CdNC_CdEC_NS.

**Minor Comments**

**1) Line 20. Isn't surface radiative flux also highly important???**

Thank you for the comment. We included the radiative flux, and the new sentence at line 19 was modified in: *"These transfers of energy are primarily driven by surface radiative flux and turbulent air–sea fluxes (TASFs), which include wind stress and the turbulent heat flux components (THFs, latent and sensible heat fluxes). "*

**2) Line 78. Re "Marsaleix et al." – based on the title of this paper, it does not obviously mention TKE, but it does mention energetics. Please confirm it is the correct reference. Sorry, I have not read it.**

Following this suggestion, we checked the references and the correct one is indeed Blanke and Delecluse, 1993. We modified the manuscript accordingly.

**3) At this point, the reviewer might anticipate experiments to look at the effect of including surface currents in stress. Your paper does not do this, which is fine, but you may want to refer to the extensive literature on the subject (e.g. Renault et al., Sun et al, and many others).**

We thank the reviewer for this comment. As requested by Referee#3, we performed an extra experiment, 1 year long, in which we applied the relative wind, instead of absolute wind, in the

ECMWF_S bulk parameterization. We refer to the new experiment as ECMWF_REL . Plot 3 presents the results. As expected, the wind stress is reduced by the inclusion of the surface ocean velocity in the bulk formula, with respect to the absolute wind simulation: the wind speed in ECMWF_REL is weaker (up to -0.2 m/s) than ECMWF_S in the equatorial band (Plot 3b). As expected from the dependencies between $C_D$ and the wind speed (Figure 1b of the manuscript), we find higher values of $C_D$ in ECMWF_REL in the area of calm wind conditions and weaker values elsewhere. Differences of $C_D$ and U between experiments are reflected onto the resulting wind stress field after bulk calculation (Plot 3c): the ECMWF_REL wind stress is weaker than ECMWF_S, especially where the U differences are higher (e.g. equatorial band). This difference in wind stress also leads to the SST differences (Plot 3d), hence ECMWF_REL results are warmer than ECMWF almost everywhere. Changes in wind stress also affect the current (Plot 3e): due to the weaker wind stress along the equator, the ECMWF_REL zonal currents are weaker than ECMWF ones. Even though the results provide insight into the effects tha bulk modifications can have in the upper ocean , we think that the current-stress negative feedback needs more and longer experiments (i.e. one for each bulk parameterization) to be properly assessed. We do not include a proper analysis in the manuscript, but we consider the effect of relative vs. absolute wind in the manuscript. Text was added in section 2.2 (lines 122-126): " *The effect of the ocean current interaction/feedback in the bulk formulation has been widely explored in the literature (e.g. Renault et al., 2019a, b; Sun et al., 2019). Although many previous studies highlighted the substantial difference in the surface input to the ocean between calculations that use absolute vs. relative wind, we have preferred to leave this aspect to further work since the implementation of this correction does substantially depend on the characteristics of the forcing fields (Renault et al., 2020).*"

[Figure]

Plot 3: Annual mean differences of a) drag coefficient ($C_D$), b) wind speed (U), c) Wind stress, d) SST and e) zonal current between ECMWF_S and ECMWF_REL.

**4) List starting Line 110. I would add:**

**3. Effect of including ocean current in stress**

**4. The form of the exchange coefficients**

**Then you can mention which of these effects you look at. Am I correct in thinking you do not explicitly look at the effect of convective gustiness? See comment later.**

We added the two bullet points and the following text (lines 121-122): "*In this study, we attempt to disentangle the effects of the first two aspects on SST (section 3.2, 3.3 and 3.4), and we discuss the effect of the inclusion of convective gustiness in the wind stress computation (section 3.4).*"

**5) Line 128 . NCAR scheme ... minimum wind speed of 0.5 m/s ... This is interesting, and I just confirmed this is also done in the CESM scheme. Note that the Large and Yeager drag coefficient actually goes to infinity as you approach zero wind speed (your Fig. 1b). So even if the wind speed gets very low, the momentum flux remains significant.**

Yes, we confirm that the description here refers to the NCAR formulation as introduced in NEMO where the minimum wind speed is used.

**6) Lines 129-134. I would say that the Large and Yeager scheme also uses MOST. It combines MOST theory with a semi-empirical form of drag coefficient.**

We clarified this point in the text. The sentence has been modified (lines 144-145): "*The NCAR parameterization uses a combination of the MOST theory with a semi-empirical form of drag coefficient in which the BTCs are computed as function of …*"

**7) Fig. 1b. I understand that you do not focus on high/extreme wind speeds, but I am curious to know what happens above 35m/s. There is some discussion on this topic in Fu et al. (2021), their sections 3.2 and 4.1. (Note that their paper employs the original Large and Yeager (2004) form of drag coefficient, without reduction at high wind speeds.) Note also that ERA5 is a high-resolution dataset and will include extreme wind events. Reference: Fu, Dan et al. 2021: Introducing the new Regional Community Earth System Model, R-CESM. B. Amer. Meteor. Soc., 102, E1821-E1843, https://doi.org/10.1175/BAMS-D-20- 0024.1**

**Lines 134-145. It would be useful to show a zoomed-in plot of Fig .1b for winds 10m/s or less.**

We thank the reviewer for suggesting these interesting papers. Here is how neutral drag and moisture transfer coefficients (thick and thin lines, respectively) vary for wind stronger than 35 m/s:

[Figure]

Plot 4: Neutral drag and moisture transfer coefficients for COARE_S (black), NCAR (blue), and ECMWF_S (green) bulk parameterizations (thick and thin lines, respectively), as functions of the neutral wind speed at 10m

Our study e does not focus on extreme wind events, then we decided to keep  the original plot (in Figure 1 of the manuscript)in the paper with a 0-25m/s range (as suggested by Referee#2)   to which we added a zoomed-in subplot for winds lower than 10m/s. The complete Figure 1 of the revised manuscript is:

[Figure]

Plot 5: a) Annual mean of $U_{N10}$ from NCAR parameterization b) and c) Neutral drag and moisture transfer coefficients for COARE (black), NCAR (blue), and ECMWF (green) bulk parameterizations (solid and dashed lines, respectively), as functions of the neutral wind speed at 10m.

**8) Line 221. Fig. 6a shows very small QT (~1 W/m2) over most of the Globe, only small regions reach 10W/m2.**

Thank you, it was a mistake. We corrected it.

**9) Line 258-259. I believe you do not explicitly look at the sensitivity to convective gustiness parameterization. So your inferences here are solely based on the fact that CD differences are small in these regions? You can consider running an extra sensitivity experiment with convective gustiness switched off in ECMWF.**

Following this comment, we performed an extra ECMWF run where convective gustiness is switched off (it is named ECMWF_NS_NG). Plot 5a compares the wind stress in the "original" ECMWF_NS and the new ECMWF_NS_NG. Results confirm that wind stress is stronger almost everywhere when the convective gustiness is included in the U calculation. Nevertheless, it is worth underling that the differences in $C_D$ between experiments are highly variable in time and that could hide the relationship between U, $C_D$ and $\tau$.

Following also suggestion by the Referee#2, we calculated the RMSE of the differences in the wind speed (U) and drag coefficient ($C_D$) between the ECMWF_NS and CdNC_CeEC_NS experiments (compare Plot 7a below with Figure 9a,c). The spatial distribution of the RMSE of U (Plot 7a) resembles the annual mean differences of U between the two experiments (shown in Figure 9a). The RMSE of $C_D$ (Plot 7b) is large where the annual mean differences between ECMWF_NS and CdNC_CeEC_NS are negative (Figure 9b,c).

For this reason, we computed the correlation between the $C_D$ differences and the $\tau$ differences for the ECMWF_NS and CdNC_CeEC_NS runs. The correlation is always significant with positive values, (Plot 6b). The higher the difference in $C_D$, the stronger the difference in wind stress.

The following text was added in manuscript (lines 289-295): "*In regions where the differences in* $C_D$ *and wind stress are opposite (e.g. the north-west and south-west Pacific and Atlantic ocean, Indian ocean, Baja California), the high time-variability of the* $C_D$ *differences (not shown) could hide the relation between* $C_D$ *and* $\tau$. *In addition, including the convective gustiness in U calculation strengthens the wind stress in ECMWF_NS. Both hypotheses are verified, the* ECMWF_NS *experiment presents a stronger wind stress almost everywhere over the global ocean compared to a twin experiment where the convective gustiness is not used in the computation (Figure S2) and the correlation between* $C_D$ *differences and wind stress differences is always significant and positive (not shown). The higher the difference in* $C_D$, *the stronger the differences in wind stress.*"

[Figure]

a) $\tau$

b) Corr Cd- $\tau$ differences (ECMWF_NS − CdNC_CeEC_NS)

Plot 6: a) Annual mean differences of wind stress between ECMWF_NS and ECMWF_NS_NG, b) correlation between $C_D$ differences and $\tau$ differences between ECMWF_NS and CdNC_CdEC_NS. Hatching indicates significant values (95% confidence level).

[Figure]

a) RMSE U between ECMWF_NS and CdNC_CeEC_NS

b) RMSE $C_D$ between ECMWF_NS and CdNC_CeEC_NS

Plot 7: Root Mean Square Error of a) wind speed (U) b) drag coefficient ($C_D$) differences between *ECMWF_NS and CdNC_CeEC_NS.*

**10) Lines 259 onwards. It is not obvious to me from Fig. 8 that the EBUS will be notable regions of enhanced stress and WSC. Perhaps zoom in on an example EBUS and show the causal links more clearly between U, CD, TAU and WSC.**

We zoomed the results over the Benguela upwelling system (Plot 8). The ECMWF_NS experiment shows a notable increase of wind stress and wind curl along the Benguela coast. As previously commented, the wind stress is not affected by the type of first-order feedback at play for the NSHFs (SST-QT negative feedback in section 2.2). The wind stress is stronger when the $C_D$ is larger in ECMWF_NS than CdNC_CeEC_NS $C_D$. The wind stress is slightly weaker when the $C_D$ is lower in CdEC_CeEC_NS than CdNC_CeEC_NS.

It is worth noting that these cross-shore differences of wind stress lead to stronger wind stress curl in ECMWF_NS with respect to CdNC_CeEC_NS.

[Figure]

Plot 8: a) Annual mean differences of a) wind speed (U), b) wind stress ($\tau$), c) drag coefficient ($C_D$), d) wind stress curl (WSC) between ECMWF_NS and CdNC_CeEC_NS. Hatching indicates significant values (95% confidence level).

[Figure]

Plot 9: a) Annual mean differences of SST between ECMWF_NS and CdNC_CeEC_NS; b) correlation between SST differences and wind stress differences between ECMWF_NS and CdNC_CeEC_NS; c) same as in b) but for SST differences and wind stress curl differences. Hatching indicates significant values (95% confidence level).

To show the relationship between variables, we show the correlation of SST differences with wind stress and wind stress curl differences (Plot 9b,c). We added Plot 9 in the supplementary material as Figure S5, and we modified Section 3.4 including the following text (lines 311-312): "…, the enhanced wind stress and negative wind stress curl in ECMWF_NS reinforce the vertical velocity with respect to CdNC_CeEC_NS (Figure S4), resulting in colder surface temperature (see correlation maps Figure S5)."

**10) On this topic, the lead author has 2 nice papers on EBUS in JRA55do and ERA-Interim-forced runs. Based on this experience, can you comment on whether the changes to TAU and WSC are realistic and whether they would make a sizable change to upwelling?**

We thank the referee for this comment. Comparing Figure 4 of Bonino et al. 2018 and Plot 10, we can notice that, during upwelling season (ONDJ), the differences in wind stress and wind stress curl between JRA55do and ERA-Interim experiments show higher range of values with respect to the wind stress and wind stress curl differences between experiment that used different $C_D$ parameterization. Since in the former pair of experiments, the atmospheric forcing is totally different, I would say that this result is quite expected. To better quantify the differences in the upwelling regions, we plotted the vertical velocity at 30m over the Benguela region (as Figure 6 in Bonino et al. 2018) for the ECMWF_NS and CdNC_CeEC_NS experiments (Plot 10c, d) and the differences between the two (Plot 9c) during upwelling season (ONDJ).

[Figure]

Plot 10: Seasonal mean differences (ONDJ) of a) wind stress (τ) and b) wind stress curl (WSC) between ECMWF_NS and CdNC_CeEC_NS; c) Differences in vertical velocity at 30m (W 30m) between the two ECMWF_NS and CdNC_CeEC_NS. Hatching indicates significant values (95% confidence level). Red square identifies the area shown in panel c).

As expected, the vertical velocity is stronger in ECMWF_NS experiments. The ECMWF_NS upwelling increases by about 30%. Comparing Figure 6 (bottom row) in Bonino et al 2018 with Plot 10, we can notice that the differences in vertical velocity are, in this study, half of the differences in Bonino et al. 2018, in terms of absolute values. It is worth noting that here the differences in wind stress and the WSC are both upwelling favorable for ECMWF_NS, while in Bonino et al 2018 wind stress differences are upwelling favorable for JRA55do and the WSC difference are upwelling favorable for ERA-Interim. Nevertheless, as expected, the differences in the wind forcing between experiments drives the difference in vertical velocity: the weaker - in terms of absolute values - the ECMWF_NS and CdNC_CeEC_NS wind forcing differences with respect to JRA55do and ERA-Interim, the weaker are the differences in vertical velocity. This suggests that the differences in wind stress and wind stress curl are comparable with results from Bonino et al 2018: the weaker (greater) the differences in wind forcing, the weaker (greater) are the differences in vertical velocity.

Plot 10 is included in the manuscript in the supplementary material as FigureS5. The following sentence was added in the manuscript (lines 309-313): *"These relations are confirmed along the coast of the Benguela Upwelling System (Figures S4 and S5). During the Benguela upwelling season310 (ONDJ), the enhanced wind stress and negative wind stress curl in ECMWF_NS reinforce the vertical velocity with respect to CdNC_CeEC_NS (Figure S4), resulting in colder surface temperature (see correlation maps Figure S5)."*

**11) Line 268. Can you see any changes to the North Equatorial Undercurrent, which is mainly an WSC-driven system (e.g. Sun et al. 2021 and references therein, https://doi.org/10.1016/j.ocemod.2021.101876 )**

We thank the reviewer for the comment and for suggesting this interesting paper. We calculated the upper 400m vertically integrated zonal currents as Sun et al. 2021. Plot 11 shows the results for ECMWF_NS and CdNC_CdEC_NS experiments, while Plot 12 shows their differences. We noticed differences in the Equatorial Undercurrent more than in the North Equatorial Countercurrent. Kessler et al. (2003) suggested that the strip of positive WSC at the equator is the key to produce the north branch of Equatorial Undercurrent and, indeed, ECMWF_S shows remarkable difference in it with respect to NCAR (see Plot 12 below). This result is certainly interesting, nevertheless, we think that a deeper analysis on the equatorial currents would require further work and it would be something

interesting to further investigate it in a future study. The current manuscript is now dense with new information and we prefer to not include more material for ease of reading.

**The upper 400 m vertically integrated zonal currents**

[Figure]

Plot 11: Upper 400m vertically integrated zonal currents for a) *ECMWF_NS  and b) CdNC_CeEC_NS.* Hatches display significant values (95% confidence level).

**a) Upper 400 m vertically integrated zonal currents**

[Figure]

Plot 12: a) Annual mean difference of Upper 400m vertically integrated zonal currents between *ECMWF_NS  and CdNC_CeEC_NS. Hatching indicates significant values (95% confidence level).*

**12) Line 286. But the equator to 10deg. N difference is still large with COARE (Fig. 2a, right)**

We thank the referee for the comment. We plotted the curl differences between the experiments (Plot 13), as also suggested by the Referee#2. This plot is added as Figure 3c in the revised manuscript. Results show that  the wind stress curl is stronger in COARE_S than in NCAR  in the north equatorial region, but this positive difference is less pronounced than ECMWF - NCAR case. We modified the paragraph in the manuscript (lines 334-338) as follows: "*As regard the equatorial upwelling, the weak increasing of the wind stress in the north equatorial region (e.g. northern equatorial cold front, Figure 3b) compared to NCAR wind stress (Figure 3a), prevents the enhancement of the positive wind stress curl in COARE_S (Figure 3c). Nevertheless, to properly identify the drivers of the pattern in the SST differences between COARE_S and NCAR extra dedicated numerical experiments should be performed.*"

c) WSC

[Figure]

Plot 13: wind stress curl (W SC) between ECMWF_S and NCAR experiments (left) and COARE_S and NCAR experiments (right). Hatching indicates significant values (95% confidence level).

**REFERENCES**

Blanke, B., & Delecluse, P. (1993). Variability of the tropical Atlantic Ocean simulated by a general circulation model with two different mixed-layer physics. *Journal of Physical Oceanography*, *23*(7), 1363-1388.

Bonino, G., Masina, S., Iovino, D., Storto, A., & Tsujino, H. (2019). Eastern Boundary Upwelling Systems response to different atmospheric forcing in a global eddy-permitting ocean model. *Journal of Marine Systems*, *197*, 103178.

Merchant, C.J., Embury, O., Bulgin, C.E., Block, T., Corlett, G., Fiedler, E., Good, S.A., Mittaz, J., Rayner, N., Berry, D. and Eastwood, S., 2019. Satellite-based time-series of sea-surface temperature since 1981 for climate applications. Nat. Sci. Data. 6:223. doi: 10.1038/s41597-019-0236-x

Kessler WS, Johnson GC, Moore DW (2003) Sverdrup and nonlinear dynamics of the Pacific equatorial currents. J Phys Oceanogr 33: 994–1008

---

## Author Comment (AC4)

**REFEREE #3**

Dear Referee,

we would like to thank you for the careful reading of the manuscript and the constructive comments that substantially helped to improve and clarify the paper. Answers to all your comments are detailed hereafter. Corrections to the English grammar were adopted in the revised version of the manuscript according to the reviewer's recommendations, but are not reported or discussed here. All authors agree with the modifications made to the manuscript. The comments by the referee are reported in bold followed by our response (in blue). The text added to the revised manuscript is reported in italic font. The revised manuscript that includes track changes and line numbers is provided in pdf format.

In the following answers, we use 'Figure' to identify the figures in the updated manuscript and we use 'Plot' to identify the figures in this document.

The name of the experiments have been slightly modified, as reported in Table 1. They are used in the following answers and in the updated manuscript.

| Experiment name | sea surface temperature used ($T_s$) | computation of $C_D$ | computation of $C_E$ and $C_H$ | convective gustiness |
|---|---|---|---|---|
| COARE_S | SSTskin | COARE3.6 | COARE3.6 | Yes |
| ECMWF_S | SSTskin | ECMWF | ECMWF | Yes |
| NCAR | SST | NCAR | NCAR | No |
| ECMWF_NS | SST | ECMWF | ECMWF | Yes |
| CdNC_CeEC_NS | SST | NCAR | ECMWF | No |
| ECMWF_NS_NG | SST | ECMWF | ECMWF | No |

**Table 1.** Summary of the numerical experiments.

**Major comments**

**1) A first important caveat of this study is the duration of the simulations, and consequently the significance of the results presented here. Even if the SST adjusts quickly and locally to surface turbulent fluxes modifications (from a few hours to a few days), the large-scale patterns and differences presented here might need more than one year to spin up and reach an new equilibrium state. The simulated interannual variability can also be modified between the sensitivity experiments, which can be misleading when interpretating the simulation differences. Another less important consequence is that spatial figures are quite noisy, which make them less readable.**

**Hence I would suggest to extend the different simulations to at least a 5-year period to make the results presented here more robust. As a comparison, Brodeau et al 2017 simulations which are referred in this manuscript discussions cover a 30-year period. If it not possible to extend the simulations for practical/technical reasons, I recommend to extend at least one simulation and compare the simulated turbulent fluxes and SST between the 1-year and the 5-year simulations to make this study more convincing.**

We agree with the referee, and we extended the ECMWF_S  and  NCAR control experiments  (i.e. experiments which do not include modification in the bulk parameterization) to 5-year simulations to assess if the SST differences noticed with 1-year simulations are robust. Model results after 5 years confirm the presence of colder SST  at the equator and over the  EBUS in CdNC_CeNC_NS (Plot 1).  We added the following text in the manuscript (lines 211-212): "*This spatial pattern of SST differences persists when extending the simulations up to 5 years (not shown).*"

a) 5 years SST differences

Plot 1: Differences of the 5 year mean SST  between ECMWF_S and NCAR  experiments. Hatching indicates significant values (95% confidence level).

**2) Another important issue is that the manuscript does not contain any validation of the simulated oceanic state, and especially the SST. I understand that a detailed validation is out of the scope of this study, but SST is the only assessed oceanic variable here, and because it is a key variable in STHF computation, we should know about the potential model biases compared to observations, and how these SST biases can modify STHF estimates (through air- sea temperature and humidity differences), and more importantly the turbulent fluxes sensitivity. SST is a well observed variable, especially at global scale and over large time period, so it would not require too much work to include an observed SST map over the same period as a reference. The idea here is not to classify the "best" bulk parameterizations, but to have a global idea of model SST biases.**

We agree with the referee. The manuscript can benefit from the  evaluation against observation, so we included some in the revised manuscript. We compared SST from the "control experiments"

against the *European Space Agency (ESA) Climate Change Initiative (CCI) SST dataset v2.0 (*ESA CCI SST, Merchant et al. 2019). Results are presented in Plot 2 which was added as Figure 2 to the revised manuscript. Text has been added from lines 193 to 201: *"We compare the SST simulated by the ECMWF_S, COARE_S and NCAR control experiments with the European Space Agency (ESA) Climate Change Initiative (CCI) SST dataset v2.0 (hereinafter ESA CCI SST dataset) which consists of daily-averaged global maps of SST on a 0.05° x 0.05° regular grid, covering the period from September 1981 to December 2016 (Merchant et al., 2019). All the control experiments present a warm bias in the Eastern Pacific, in the Eastern Boundary Upwelling systems (EBUS), in the Western Boundary Currents (WBCs) and in the Antarctic Circumpolar Current (ACC) region. The SST reproduced by COARE_S and ECMWF_S shows a cold bias of about -1°C in the North Atlantic open ocean at mid- latitudes, and a warm bias of about 0.5°C in the Indian Ocean and the Western Pacific (Figure 2a,b); NCAR SST is also colder than observations, with a larger bias of about -2°C in the North Atlantic (Figure 2c). The bias is generally higher compared with other two experiments and covers wider areas. "*

[Figure]

Plot 2: Annual mean SST differences between a) ECMWF_S, b) COARE_S, and c)NCAR against ESA CCI SST.

**3) My last major point concerns the surface current effect into the surface stress computation. Considering absolute or relative winds in stress formula in forced ocean simulation is still debated, but an additional sensitivity experiment using relative wind could give additional insight (as it is done for wind gustiness for example) to this manuscript compared to Brodeau et al. 2017. From my understanding, as the prognostic SST does not influence surface stress (or very weakly through stability functions), your sensitivity experiments using different Cd is totally similar to Brodeau et al. 2017, and hence leads to the same already-known results. This additional experiment would allow to assess the current-stress negative feedback (as it is done for SST-STHF feedback in 3.5), and how it changes the stress sensitivity to the bulk choice. This would substantially enrich the 3.4 section of the manuscript, which is currently of limited interest.**

We thank the referee for this interesting comment. We performed an extra experiment, 1 year long, in which we applied  the relative wind, instead of absolute wind, in the ECMWF_S bulk parameterization. We refer to the new experiment as ECMWF_REL . Plot 3 presents the results. As expected, the wind stress is reduced by the inclusion of the surface ocean velocity in the bulk formula, with respect to the absolute wind simulation: the wind speed in  ECMWF_REL is weaker (up to -0.2 m/s) than ECMWF_S in the equatorial band (Plot 3b). As expected from the dependencies between $C_D$ and the wind speed (Figure 1b of the manuscript), we find higher values of $C_D$ in ECMWF_REL in the area of calm wind conditions and weaker values elsewhere. Differences of $C_D$ and U between experiments are reflected onto the resulting wind stress field after bulk calculation (Plot 3c): the ECMWF_REL wind stress is weaker than ECMWF_S, especially where the U differences are higher (e.g. equatorial band). This difference in wind stress also leads to the SST differences (Plot 3d), hence ECMWF_REL results are warmer than ECMWF almost everywhere. Changes  in wind stress also affect the current (Plot 3e): due to the weaker wind stress along the equator, the ECMWF_REL zonal currents are weaker than ECMWF ones. Even though the results provide insight into the effects tha bulk modifications can have in the upper ocean , we think that the current-stress negative feedback needs more and longer experiments (i.e. one for each bulk parameterization) to be properly assessed. We do not include a proper analysis in the manuscript, but we consider the effect of relative vs. absolute wind in the manuscript. Text was added in section 2. (lines 122-126): ” *The effect of the ocean current interaction/feedback in the bulk formulation has been widely explored in the literature (e.g. Renault et al., 2019a, b; Sun et al., 2019). Although many previous studies highlighted the substantial difference in the surface input to the ocean between calculations that use absolute vs. relative wind, we have preferred to leave this aspect to further work since the implementation of this correction does substantially depend on the characteristics of the forcing fields (Renault et al., 2020).*”

[Figure]

Plot 3: Annual mean differences of a) drag coefficient ($C_D$), b) wind speed (U), c) Wind stress, d) SST and e) zonal current between ECMWF_S and ECMWF_REL.

**Minor comments**

**1) A lot of English typos can be found in the text. A careful check is needed. Some punctuations are also missing.**

**Spatial figures must be improved to reach publication quality requirements. Here is some recommendations to improve them:**

**Spatial figures color extremes are often too much saturated and iso-contours are too thick. They are also very noisy due to the short experiments length. All these aspects make them hardly readable. Latitudes should also be extended from 70°S to 70°N as in Brodeau et al. to facilitate the comparison between those 2 studies.**

**A longitudinal average would also greatly improve and simplify figures interpretation as most of the results are mainly latitude dependant.**

**Some figures have resolution issue and appears blurry when zoomed in.**

We agreed with the referee and we carefully checked the text and the punctuation. We also reproduced all the figures at higher resolution, and  extended the latitudinal range from 70°S to 70°N. Palettes, saturation and contours were modified following the referee comments. New figures are clearer and it is  easier to interpret. We preferred to maintain the lat-lon maps which show the spatial variability.

REFERENCES:

Merchant, C. J., Embury, O., Bulgin, C. E., Block, T., Corlett, G. K., Fiedler, E., Good, S. A., Mittaz, J., Rayner, N. A., Berry, D., et al.: Satellite-based time-series of sea-surface temperature since 1981 for climate applications, Scientific data, 6, 1–18, 2019.

---

## Author Comment (AC5)

**REFEREE #2**

Dear Referee,

we would like to thank you for the careful reading of the manuscript and the constructive comments that substantially helped to improve and clarify the paper. Answers to all your comments are detailed hereafter. Corrections to the English grammar were adopted in the revised version of the manuscript according to the reviewer's recommendations, but are not reported or discussed here. All authors agree with the modifications made to the manuscript. The comments by the referee are reported in bold followed by our response (in blue). The text added to the revised manuscript is reported in italic font. The revised manuscript that includes track changes and line numbers is provided in pdf format.

In the following answers, we use 'Figure' to identify the figures in the updated manuscript and we use 'Plot' to identify the figures in this document.

The name of the experiments have been slightly modified, as reported in Table 1. They are used in the following answers and in the updated manuscript. We tried the names that you suggested, but they were hard to read throughout the text.

| Experiment name | sea surface temperature used ($T_s$) | computation of $C_D$ | computation of $C_E$ and $C_H$ | convective gustiness |
|---|---|---|---|---|
| COARE_S | SSTskin | COARE3.6 | COARE3.6 | Yes |
| ECMWF_S | SSTskin | ECMWF | ECMWF | Yes |
| NCAR | SST | NCAR | NCAR | No |
| ECMWF_NS | SST | ECMWF | ECMWF | Yes |
| CdNC_CeEC_NS | SST | NCAR | ECMWF | No |
| ECMWF_NS_NG | SST | ECMWF | ECMWF | No |

**Table 1.** Summary of the numerical experiments.

**Major comments**

**1) A lot of effort is put in explaining quite successfully why there is a cold bias in the equatorial eastern Pacific and the EBUS regions between ECMWF and NCAR bulk schemes. However, very little attention is given to the strong differences in heat fluxes found over western boundary current (WBC) systems, which are known to be areas with strong air-sea interactions (See figure 2 of the manuscript). I invite the authors to dig a bit more in this direction and explore if there is a meaning in the spatial and temporal variability of this difference. If it was only noise, I would expect it to average to zero in an annual mean, but, indeed, it is visible in both ECMWF-NCAR and COARE-NCAR differences (figure 3).**

We agree with the referee that this study could benefit from a detailed analysis of the WBC systems , given the strong air-sea interactions that characterize them. Therefore, we analyze the total turbulent heat fluxes in the WBC region to highlight the effect of employing a different THF computation in CdNC_CeEC_NS and NCAR. To better understand the spatial variability over the WBCs regions, we focus on the differences in the surface sea temperature (SST) and total turbulent fluxes (QT) over the Gulf Stream and Kuroshio current areas as shown in Plot 1. Since the spatial patterns are quite heterogeneous due to mesoscale activities, we study the relationship between QT and the air-sea virtual temperature difference for the winter (DJF) and summer (JJA) seasons. We selected specific regions along the currents path (identified by the yellow squares in the following Plot1) . Plot 2 (middle and bottom rows) show the relationship for the grid points inside the yellow squares in Plot2 (top row).

Notably, the two experiments behave in a different way along the WBC. In both seasons, the relationship between QT and the air-sea virtual temperature difference in CdNC_CeEC_NS is clearly shifted with respect to NCAR. When the temperature differences are negative $(T_VSEA>T_VAIR)$, CdNC_CeEC_NS shows lower temperature differences in terms of magnitude compared to NCAR. These results are consistent with the finding in the "Turbulent Heat Fluxes" section  (i.e. 3.3 section) of the revised manuscript.

Plot 2 is included in the supplementary material as Figure S1 and the following text describes it in Section 3.3 (lines 354-359): "…, the CE of CdNC_CeEC_NS, which is smaller than CE of NCAR (Figure 8a), induces weak evaporation. The resulting weaker heat loss to the atmosphere in CdNC_CeEC_NS with respect to NCAR implies a gain of heat by the ocean (positive regions in Figure 7a) of about 2W/m2 over low-latitudes and up to 6 W/m2 over mid-latitudes (Figure 7b). A similar process is acting also in areas where the annual mean pattern of QT is patchy due to the mesoscale activities in both in summer and winter seasons (e.g. in the Western Boundary Currents, Figure S1). In CdNC_CeEC_NS, the negative virtual temperature differences at the air-sea interface are smaller than NCAR, inducing weaker from the ocean to the atmosphere "

[Figure]

Plot 1: Annual mean differences of a) SST, b) QT between CdNC_CeEC_NS and NCAR for Kuroshio current (left column) and Gulf stream (right column). Hatching indicates significant values (95% confidence level).

[Figure]

Plot 2: Zoom of annual mean differences of total turbulent heat fluxes (QT) between CdNC_CeEC_NS and NCAR experiments over Gulf Stream and Kuroshio current (top row); Relationship between total turbulent fluxes (QT) and the air-sea virtual temperature difference for selected grid points inside the yellow squares in CdNC_CeEC_NS (blue circles) and NCAR (red circles) in winter (middle row) and in summer (bottom row) for Gulf Stream and Kuroshio current.

**2) What are the limitations in running simulations that last one year only? Is there any dependence on the specific year (e.g. in terms of ENSO phase, or any other climatic mode)? What about the spinup of the model? Which year has been considered?**

We thank the referee for the comment and we apologize for the missing information in the manuscript. All the results presented in the manuscript are based on 1-year simulations (2016) with 1 previous year of spinup (i.e. 2015). The length of the simulations is long enough to identify the short-term impact of changes in the bulk parameterization on the upper ocean characteristics. We clarified this in the revised manuscript (lines 160-161): "All the experiments are 1-year long experiments, starting from January 2016 after 1-year of spinup." In order to prove the robustness of the SST differences simulated in the 1-year experiments , we extended ECMWF_S and NCAR experiments (following also suggestions by the Referee #3). Model results of 5 year long runs confirm that CdNC_CeNC_NS simulates colder SST at the equator and over the EBUS in CdNC_CeNC_NS (Plot 3). The following sentence was added (lines 211-212): "*This spatial pattern of SST differences persists when extending the simulations up to 5 years (not shown).*"

a) 5 years SST differences

[Figure]

Plot 3: Differences of the 5 year mean SST between ECMWF_S and NCAR experiments. Hatching indicates significant values (95% confidence level).

**2) With respect to Brodeau et al. (2017, B17 hereafter), the fluxes are computed using a dynamical SST field that responds to the atmospheric forcing. However, the atmospheric dynamics is known to respond to the SST even on daily and sub-daily time scales (see the review of Small et al., 2008, and some examples of applications in different areas of the world such as Li and Carbone, 2012; Gaube et al., 2019; Desbiolles et al., 2021). It would be interesting to discuss a fully coupled approach, as it has been shown that surface winds and clouds are affected by the SST structure on daily time-scales which, then, affect the SST and the surface turbulent fluxes back. This is only mentioned at the end of the manuscript and it should probably be included in the Introduction, as well. Moreover, the closed loop of this kind of ocean-atmosphere interactions has been proposed to be responsible for a three-to-six day oscillation (Strobach et al., 2020): I wonder if these oscillations are also observed here and whether they depend on the flux parameterizations.**

In accordance with the Referee's suggestion, we added a paragraph in the introduction about the fully coupled approach. We do agree that analyzing the impact of a modified SST on the atmosphere properties, i.e. winds and clouds, in a coupled system would be interesting. Nevertheless, the scope of this study is to investigate the role of the bulk formulations in a bulk-forced OGCM. This approach is still largely used by the ocean modeling community in a variety of applications.

As regards the three-to-six day oscillation, unfortunately we cannot analyze such high-frequency variability  because the model output of our simulations was saved as a 5-day average. The new text in the introduction (lines 39-46) reads as : *"It is worth mentioning that the online prognostic approach does only partially close the air-sea feedback. Surface winds and clouds are affected by the SST structure on daily time-scale which, in turn, affect the SST and the TASFs (Desbiolles et al., 2021; de Szoeke et al., 2021; Gaube et al., 2019; Li and Carbone, 2012; Small et al., 2008). The closed air-sea feedback (hereinafter coupled approach) in the system might substantially impact the turbulent fluxes (Lemarié et al., 2021; Small et al., 2008), but the coupled approach is still not yet mature in the ocean model community. Recently Lemarié et al. (2021) implemented a first attempt of a simplified atmospheric boundary layer model (ABL) to improve the representation of air-sea interactions in*

*NEMOv4.2. However, the online prognostic SST approach is still largely used by the ocean modeling community in a variety of applications."*

**3) In general, the fact that full ocean simulations are performed seems a bit underexploited.**

We agree with the referee that the oceanic responses might be more deeply exploited. Following also a comment from Reviewer#1, we extended the analysis on how modifications in the bulk parameterization influence the poleward transport of heat in the upper ocean. We computed it in the upper 100m of the global ocean and analyzed the differences among experiments (in Plot 4). The MHT in the ECMWF experiments is always higher compared to experiments that employ $C_D$ from NCAR formulation. ECMWF (with and without skin temperature) and NCAR-based experiments present the largest differences (Plot 4 a,e) that peak generally in the tropical band (about 0.8 PW, 20% of NCAR absolute value) where ECMWF wind stress is stronger than NCAR one. Global MHT in COARE_S and NCAR runs is comparable (Plot 4b), with differences lower than 0.3 PW. The transport differs only in the tropical latitude band in all the experiments that used the same $C_D$ (i.e. ECMWF_S and ECMWF_NS; NCAR and CdNC_CeEC_NS), and they are quite small (about 0.1 PW, Plot 4 c,d).

Plot 4 (a,b) has been included in Section 3.3 of the revised manuscript, as Figure 4c,d; Plot 4e is in Section 3.4 as Figure 11c.

The following text was added to describe Figure 4 (lines 219-225): *"Changes on the simulated SST can reflect on the temperature profile in the upper ocean and the distribution of heat on global scales. We have computed the global ocean heat transport in the upper 100 meters and compared it among experiments. Figure 4 (c,d) presents the meridional heat transport (MHT) as a function of latitude. The MHT is larger in ECMWF_S compared to NCAR mostly at all latitudes (Figure 4c), with the largest differences (about 0.8 PW, 20% of NCAR absolute value) in the tropical band where ECMWF wind stress is stronger than NCAR one (Figure 3a). COARE_S and NCAR compare well, with differences lower than 0.3 PW (Figure 4d). Then, we will focus only on the differences between ECMWF_S and NCAR to analyze in detail the relationship between TASFs and SST. We show differences in MHT only when relevant."*

The following text added in Section 3.4 (lines 313-317) describes Figure 11c: *" It is important to highlight that the differences in the wind stress are also responsible for the changes in the meridional heat transport. MHT differences between ECMWF_NS and CdNC_CeEC_NS resemble the differences between ECMWF_S and NCAR (compare Figure 4c and Figure 11c), with a higher transport in ECMWF_NS at all latitudes. The largest differences are located in the tropical region (up to 0.6 PW, about 18% of NCAR mean value), where the differences in meridional transport (linked to the equatorial upwelling) between the two experiments are likely maxima".*

In addition, we included the following text in the conclusion section (lines 384-385): *"Stronger wind stress results in an increase of the poleward heat transport in the upper ocean, which a more pronounced increase in the ±20 latitude band."*

[Figure]

Plot 4: Global Meridional heat transport in the upper 100m ocean (values on the right y axis) and differences (values on the left y axis) between a) ECMF_S and NCAR, b) COARE_S and NCAR, c)ECMWF_S and ECMWF_NS, d) CdNC_CdEC_NS and NCAR, and e) ECMWF_NS and CdNC_CdEC_NS.

**I think that much more information could be extracted, for example when discussing the role of different wind stress and wind stress curl in controlling the surface cooling in the EBUS and equatorial regions. Would it be possible to disentangle the role of upwelling and the role of entrainment in this surface cooling? What about doing some heat budget in the oceanic mixed layer to understand what processes are mostly modified by the different bulk algorithms?**

These simulations might be surely used for more detailed analysis on global scales and in specific regions. The analysis of the heat budget in the mixed layer might add value to our study, but it would largely expand the scope of this study. The entrainment at the base of the mixed layer is composed of three terms: the entrainment due to the vertical velocity, the entrainment due to the tendency of the mixed layer depth, the entrainment due to lateral induction of the mixed layer depth (Vijith et al 2020). If we consider only the vertical components, the vertical entrainment velocity is the sum of the second term in the equation below, which is the vertical velocity associated with the wind stress (e.g upwelling process), and the first term, which is associated with mixed layer depth (h) deepening. So that the entrainment velocity can be expressed as (Alexander, 1992; Mendoza et al., 2005):

$$ W_e = \frac{\delta(h)}{\delta(t)} + W_E \quad with \quad W_E = \frac{curl(\tau)}{\rho * f} $$

The two terms of the entrainment velocity are presented in Plot 5. As expected, the Ekman pumping velocity is positive, meaning that sea water is moving upward. The velocity associated with mixed layer depth tendency is also positive, meaning that the mixed layer deepens and promotes the entrainment of cold water from below. However, it is much weaker than the Ekman pumping velocity. Although

this is just a preliminary analysis and a complete heat budget would be necessary to verify this hypothesis, this result suggests that the surface cooling is strictly linked to the upwelling process.

[Figure]

Plot 5: Ekman pumping velocity (WE) and the mixed layer depth tendency ($\partial h/\partial t$) in the CdEC_CeEC_NS experiment.

**4) The statistical significance of the differences between the experiments should be assessed. If the distributions are Gaussian, a t-test should be enough.**

Following this suggestion, we compute the 95% significance level of the differences. We added the significance in all figures (as haches).

**5) There are various differences with respect to the estimates shown in B17. In particular:**

1. **the authors consider a single year, whereas B17 consider the period 1982-2014;**

2. **the authors use ERA5 data to force NEMO, whereas B17 use ERA-Interim data;**

3. **COARE 3.5 is used here, and COARE 3.0 is used in B17;**

4. **Different versions of the ECMWF model are considered (cycle 40 and 41).**

**For these reasons, I would be more cautious in comparing the present results with those presented in B17. Would it be possible, for example, to compute the heat fluxes using the local midnight SST throughout the day, to mimic the fixed-SST approach, as in B17, and compare these fluxes to the prognostic-SST ones? This would avoid all the limitations highlighted before, as the original data would be the same.**

We are fully aware of these differences and we completely agree that comparisons must be done more cautiously. We added a paragraph in section 3.5 to clearly list the discrepancies between the two setups and facilitate the interpretation of the results .

First, regarding point "4", no differences exist in the ECMWF algorithms between B17 and the present paper. The code source used for both studies originates from the AeroBulk package written by *Laurent Brodeau The*. ECMWF algorithm is based on the IFS documentation (doc and code for both cycles). *Brodeau* also introduced and ported all the bulk algorithms that he initially wrote for AeroBulk into the NEMO code, version 4.0 and onwards. We are not aware of any differences between cycle 40 and 41, and we have not introduced any in AeroBulk in anycase. We think that mentioning the IFS cycle was unnecessary and confusing since the true code in both studies originates from the AeroBulk package, so we removed the mention of IFS cycles and instead added mention of AeroBulk.

We modified the text as follows (lines 134-138): *"In this study, we focus on three bulk parameterizations implemented in NEMOv4.0: NCAR (Large and Yeager, 2009), COARE 3.6 (Edson et al. (2013) + private communication by Chris Fairall, hereinafter referred to as "COARE"), and ECMWF as coded in the Aereobulk package (Brodeau et al., 2017). All the codes to estimate TASFs in the NEMOv4.0 framework, originates from this AeroBulk package, which is completely open source and available at https://github.com/brodeau/aerobulk (Brodeau et al., 2017)."*

With respect to COARE3.0 used in B17, COARE3.5 introduces the developments of Edson *et al.*, 2013 which includes only modification on the drag coefficient $C_D$. Note that version "3.5" is also mentioned and briefly discussed in B17. The difference is shown in Plot 6 that is taken directly from the AeroBulk main page.

[Figure]

Plot 6: Neutral drag and moisture transfer coefficients for COARE 3.0 (yellow) COARE 3.6 (brown), NCAR (gray), and ECMWF (blue) bulk parameterizations (thick and thin lines, respectively), as functions of the neutral wind speed at 10m

When using COARE3.6, for a given near surface stability state, significantly stronger wind stress is expected above 15 m/s compared to COARE3.0, while slightly weaker wind stress is expected in calmer conditions, between 3 and 7 m/s. We see that "3.6" in the Figure 1a of the manuscript is the same thing as "3.5" when it comes to $C_D$ (compare Figure 1a and Plot 6). What we refer to "3.6" in AeroBulk introduces the latest improvements for $C_E$ and $C_H$ suggested by Chris Fairall (see response 4 of minor comments here below).

Finally, we think that your suggestion to recompute an "offline" version of the fluxes based on a SST extraction from the NEMO simulations would actually be an extremely thorough way to compare the "OGCM-online" and "prescribed-offline (B17)" approaches. However, it would represent a substantial amount of work that we do not think is fully justified in the present case, since the focus of our paper is more on "what to expect when choosing all these different algorithms and options to drive an OGCM with a prescribed surface atmospheric state?". We have been more cautious when comparing with B17 and we added the paragraph that lists the discrepancies between this study and that of B17.

We added the following text in Section 3.5 (lines 343-346): "*It is worth mentioning that there are few discrepancies in the bulk implementation between this study and Brodeau et al. (2017) . They used the COARE3.0 parameterization instead of COARE3.6 and, their simulations, performed for a longer (1982-2014) period, are forced by the  ERA-Interim reanalysis instead of ERA5. Therefore, our scope in this comparison is only to qualitatively understand the negative feedback between the SST and the QT at play in our experiments. *"

**Minor comments**

**1) There are typos throughout the text: I suggest a careful reading of the manuscript. Many maps are hard to read, because the contour lines often mask the color shading. I suggest:**

- **enlarging the maps (as currently done in Fig 5, at least);**
- **reducing the number of contour lines (or removing them, if not necessary);**

- **verifying that the contours are properly plotted and not broken at 180° or 0° longitude;**

- **removing the word 'exp =' in the titles, as it is redundant.**

We carefully checked the text and reproduced all the figures with higher resolution using a wider latitudinal range from 70°S to 70°N. We changed palettes, saturation and contours following the referee suggestions.

**Technical comments**
**1) L50: It is true that with the prognostic SST approach, there is a negative feedback between the heat fluxes and the SST, but having a dynamical ocean can also modify the heat fluxes in the other direction. Namely, the heat fluxes can be strengthened (in absolute value) with the upper ocean mixing. Is there a way to disentangle these two contributions?**

We thank the reviewer for this comment. The calculation of the heat budget should be performed to disentangle the contribution of the upper ocean mixing on the surface heat fluxes. Although, as previously mentioned, the analysis of the heat budget in the mixed layer might add value to our study, it would largely expand its scope and modify the focus of the paper. To compute an accurate heat budget we need higher model output frequency (higher than 5-day average) and additional variables in outputs, not saved for our experiments. To limit the time of the analysis and remain focused on objectives of this study, we analyzed the ocean vertical heat diffusivity ($m^2/s$) at the mixed layer depth as computed in the ECMWF_NS experiment (Plot 7) and show that it tends to zero over the interested area, suggesting that the ocean mixing is weak at the base of the mixed layer. However, these are not conclusive results , we do prefer not to include them into the manuscript.

**a) Ocean vertical heat diffusivity of ECMWF_NS at MLD**

[Figure]

Plot 7: Ocean vertical heat diffusivity of ECMWF_NS at the base of ocean mixed layer depth.

**2) L97: $Q_T$ is dominated by $Q_L$ because $Q_L$ is much larger than $Q_S$. I wonder if the buoyancy flux, in which the sensible and the latent heat flux terms are comparable, is a more appropriate variable to consider. A recent example of its dynamical importance is the work by De Szoeke et al. (2021), where the buoyancy flux is shown to control the low-level cloud formation in the tropical Indian Ocean. This, then, has a significant influence on the surface fluxes.**

We thank the referee for suggesting this interesting paper. We computed the buoyancy flux (Equation 1 of De Szoeke et al. 2021) instead of QT in Plot 2 (middle and bottom rows) of this document. Plot 8 shows the relationship, for the grid points inside the yellow squares in Plot 2 (top row), between buoyancy flux and the air-sea virtual temperature difference in CdNC_CeEC_NS and NCAR in winter and in summer for Gulf Stream and Kuroshio current. In both seasons, when the temperature differences are negative ($T_V SEA > T_V AIR$), CdNC_CeEC_NS shows lower temperature differences in terms of magnitude than NCAR. This difference tends to make the CdNC_CeEC_NS ocean surface less buoyant than NCAR and it promotes weaker heat loss from the ocean to the atmosphere for the CdNC_CeEC_NS. Since these results using the buoyancy flux resemble the ones using the QT, we decided to add Plot 2 in the manuscript to be coherent with the rest of the text.

Moreover, it is important to note that our bulk algorithms do not need clouds coverage as input. To evaluate the direct impact of clouds coverage on surface fluxes we would need a coupled system with a proper atmospheric component.

[Figure]

Plot 8: Relationship between buoyancy flux (about 1E-7) and the air-sea virtual temperature difference in CdNC_CeEC_NS (blue circles) and NCAR (red circles) in winter (left column) and in summer (left column) for a) Gulf Stream and b) Kuroshio current.

**3) LL106-107: The explanation of the cool-skin effect is not very clear.**

We thank the referee and we rephrased the sentence at lines 111-112 as: "*The cool skin is the cooling of the millimeter-scale uppermost layer of the ocean to ensure a steep vertical gradient of temperature which sustains the heat flux continuity between ocean and atmosphere.*"

**4) L124: Edson et al. (2013) introduced COARE 3.5 and not 3.6. This should be corrected throughout the manuscript.**

The most recent reference for the COARE formulation is actually Edson et al (2013). There is not a specific report or paper that describes the latest version 3.6. The COARE3.6 as implemented in NEMO (https://github.com/brodeau/aerobulk/blob/master/src/mod_blk_coare3p6.f90) was further modified based on private communication with C. Fairall. Correction done in the manuscript.

**5) Fig1: Instead of using thin lines for the moisture transfer coefficients, thick dashed lines would be more visible and easier to distinguish from the drag coefficients. I would also suggest reducing the range of wind speed in panel (b) up to 22 or 25 m/s, as the focus is on the left side of the panel.**

Here the new Figure 1b:

[Figure]

Plot 9: Neutral drag and moisture transfer coefficients for COARE (black), NCAR (blue), and ECMWF (green) bulk parameterizations (thick and dashed lines, respectively), as functions of the neutral wind speed at 10m

**6) LL150-163: I would suggest removing the bullet points and use plain text, instead, to remove the repetitions and enable a smoother reading. Table 1 is already giving a schematic recap of the experiment setup. It is also not clear what is the difference between the parameterizations that use the absolute wind speed (as in experiments ECMWF_S, COARE_S, ECMWF_NS and CdNCAR_CeEC) and the parameterization that does not include the current correction (as in the NCAR experiment). It seems that the ocean surface currents are never used in this set of experiments (L169). Thus, it can simply be stated once, as this is not a parameter that changes. I also find the names of the experiments very confusing: what about making them more explicit with something like: CdEC_CeEC (instead of ECMWF_S), CdCO_CeCO (COARE_S), CdNC_CeNC_NS (NCAR), CdEC_CeEC_NS (ECMWF_NS), CdNC_CeEC_NS (CdNCAR_CeEC)? In this way, the differences among them are readily available in their names.**

**Table 1: What about adding 'Experiment name' in the first row of the first column? A column indicating whether the gustiness in the computation of the wind stress is included would be useful.**

We agree with the reviewer, we removed the bullet points and we added a statement on absolute wind speed (lines 181-182): "*We use the absolute wind, e.g. all parametrizations do not include the ocean currents feedback to calculate wind in equation 1a.*"

**8) Fig4: The skin SST effect has a component at the daily scale. I wonder if, by considering the annual mean, the signal averages to zero. What about computing the temporal standard deviation of the difference SSTskin-SST?**

The annual mean is computed using 5-day mean model output. We computed the temporal standard deviation of the difference between SSTskin and SST between ECMWF_S and ECMWF_NS (Plot 10) and find that it is high in regions where the annual differences are highly patchy and not statistically significant (compare Plot 10 with Figure 5a). The annual differences are kept in the revised manuscript.

a) RMSE between SSTskin of ECMWF_S and SST of ECMWF_NS

[Figure]

Plot 10: Temporal standard deviation of the difference SSTskin-SSTdifferences between ECMWF_S and ECMWF_NS.

**9) LL202-207: The link between the figure and the text is not fully clear. By looking at the figure one might think that, on the annual average, there is an increase of SST when using the SSTskin correction (is the sign of the difference ECMWF_S-ECMWF_NS correct?), because of a dominant diurnal warming effect. This is in contrast with the statement that the cool skin effect is dominant over the warm layer one. Is the annual mean computed using hourly outputs? How is the mean warming interpreted? What about its spatial structure?**

The annual mean is computed using 5day mean model outputs. We reported an increase of the SST when the SSTskin is used in the computation of turbulent heat fluxes (Figure 5b). This is due to the fact that the SSTskin, used to compute turbulent heat fluxes in ECMWF_S, is colder than the SST, used to compute turbulent heat fluxes in NCAR (Figure 5a). The SSTskin is usually colder than SST because the cool-skin effect is dominant over the warm layer effect (Brodeau et al. 2017). The colder SSTskin in ECMWF_S with respect to ECMWF_NS SST yields a slightly weaker heat loss to the atmosphere due to the decreased NSHFs (mostly evaporation). Therefore, the resulting SST of ECMWF_S is warmer than the ECMWF_NS SST (Figure 5c).

**10) LL229-230: Is it 'higher heat absorption' or 'weaker heat loss'? The logical link between the latent heat considerations and the fact that it is the wind stress to be responsible for the observed cold SST pattern difference between ECMWF_S and NCAR is not clear.**

We thank the reviewer for the comment. It is weaker heat loss, so we modified the sentence. We agree with the reviewer, the logical link between the latent heat considerations and the fact that it is the wind stress that is responsible for the observed cold SST pattern difference between ECMWF_S and NCAR is not clear, so we removed the sentence.

**11) Fig6: Panel b) is not showing time series: the caption of the figure should be modified. From this figure one would not say that the mean excess QT is 10W/m2, as stated at line 221: where does this amount come from?**

We apologize for the mistake, we corrected it.

**12) L239: Up to now, it is not very clear which parameterizations use the gustiness correction in the computation of the wind stress. As noted above, this information could be included in Table 1 and some more details on how the gustiness is included in the scheme should be given.**

We changed Table 1 as reported at the beginning of this document.

**13) Fig8: I suspect that the gustiness correction is highly variable in time on daily or even sub-daily scale. As for the CSWL correction, thus, I am not sure that showing the annual mean of such variables is enough. Wouldn't it be of interest to show the variance or the RMSE of the two model setup to better display where this highly variable correction is relevant?**

This suggestion allowed us to better understand the relevance of the different coefficients in the bulk algorithms. We calculated the RMSE of the differences in the wind speed (U) and drag coefficient ($C_D$) between the ECMWF_NS and CdNC_CeEC_NS experiments (compare Plot 11a below with Figure 9a,c). The spatial distribution of the RMSE of U (Plot 11a) resembles the annual mean differences of U between the two experiments (shown in Figure 9a). The RMSE of $C_D$ (Plot 11b) is large where the annual mean differences between ECMWF_NS and CdNC_CeEC_NS are negative (Figure 9b,c). In these regions, the differences in $C_D$ between experiments are highly variable in time and they could hide the relationship between U, $C_D$ and $\tau$.

For this reason, we computed the correlation between the $C_D$ differences and the $\tau$ differences for the ECMWF_NS and CdNC_CeEC_NS runs. The correlation is always significant with positive values, (Plot 12b). The higher the difference in $C_D$, the stronger the difference in wind stress.

Moreover, to evaluate the effect of the convective gustiness on the wind stress (following a suggestion by the referee #1), we performed an extra ECMWF run where convective gustiness is switched off (it is named ECMWF_NS_NG). Plot 12a compares the wind stress in the "original" ECMWF_NS and the new ECMWF_NS_NG. Results confirm that wind stress is stronger almost everywhere when the convective gustiness is included in the U calculation. We added Plot12a as FigureS2 in the supplementary material.

The following text was added in manuscript (lines 289-295): "*In regions where the differences in $C_D$ and wind stress are opposite (e.g. the north-west and south-west Pacific and Atlantic ocean, Indian ocean, Baja California), the high time-variability of the $C_D$ differences (not shown) could hide the relation between $C_D$ and $\tau$. In addition, including the convective gustiness in U calculation strengthens the wind stress in ECMWF_NS. Both hypotheses are verified, the ECMWF_NS experiment presents a stronger wind stress almost everywhere over the global ocean compared to a twin experiment where the convective gustiness is not used in the computation (Figure S2) and the correlation between $C_D$*

*differences and wind stress differences is always significant and positive (not shown). The higher the difference in $C_D$, the stronger the differences in wind stress."*

[Figure]

**Plot 11:** a) Root Mean Square Error of a) wind speed (U) b) drag coefficient ($C_D$) differences between *ECMWF_NS and CdNC_CeEC_NS.*

[Figure]

**Plot 12:** a) Root Mean Square Error of a) wind speed (U) b) drag coefficient ($C_D$) differences between *ECMWF_NS  and CdNC_CeEC_NS. Hatching indicates significant values (95\% confidence level).*

Moreover, to be coherent with the previous finding, we performed the correlation between $C_E$ and latent heat differences between CdNC_CeEC_NS and NCAR (Plot 13). The correlation is always high, significant and negative. The lower the difference in $C_E$, the stronger are the differences in latent heat. This finding is coherent with our result that the QT in CdNC_CeEC_NS is higher by ~1W/m2 than NCAR due to a lower $C_E$ (i.e. the ocean in CdNC_CeEC_NS is less evaporative), which leads to higher latent heat (i.e. the ocean in CdNC_CeEC_NS gains heat).

c) Corr $C_E$ - Latent Heat differences

[Figure]

Plot 13: Correlation between $C_E$ and latent heat differences between CdNC_CeEC_NS and NCAR. Hatching indicates significant values (95% confidence level).

**14) L260: As the outputs of the ocean model are available, would it be possible to quantify the contribution of the surface cooling due to the modified Ekman suction between the configurations? Can a scaling between the anomalous wind curl and the anomalous SST cooling be derived?**

Following this suggestion, we correlated the SST and the wind stress curl differences between ECMWF_NS and CdNC_CeEC_NS (Plot 14c).

A strong negative correlation appears at the equator. High differences in the wind stress curl (positive north of the equator in Plot 14), correspond to negative SST differences between experiments. We added this figure in the supplementary material as Figure S3 and we added the following text in the manuscript(lines 302-304): "*We found this relation significant north of the equator: the stronger positive wind stress curl in ECMWF_NS than CdNC_CeEC_NS results in a colder SST in ECMWF_NS compared to CdNC_CeEC_NS (see correlation map in Figure S3).*"

[Figure]

a) SST

ECMWF_NS – CdNC_CeEC_NS

b) WSC

c) Corr SST-WSC differences

Plot 14: Annual mean differences of a) SST and b) wind stress curl (WSC) between ECMWF_NS and CdNC_CdEC_NS; b) correlation between SST WSC differences differences ECMWF_NS and CdNC_CdEC_NS. Hatching indicates significant values (95% confidence level).

**14) Figure 2 is not described in the main text, please do. What is its link with the SST bias? The name of the experiments should be kept consistent throughout the text. Here, for example, 'COARE3.6_S' should be replaced with 'COARE_S', or its correct name. It should be clarified (and motivated) whether the annual mean of the percentage difference or the percentage difference of the annual means is computed.**

**LL185-192: This paragraph is rather general and could be moved backward in the manuscript. One would expect here to find a reasoning on figures 2 and 3, such as why such patterns are observed, which specific reasons could explain them and their relationship, etc.**

**LL277-289: By looking at figure 2 one would expect stronger differences in the SST in the COARE-NCAR comparison, and not in the ECMWF-NCAR one. What about showing the mean difference of the wind stress (not in percentage) and, maybe, the mean difference in wind stress curl, as it relates to the upwelling?**

In accordance with the Referee's suggestions, we changed Figure 2 (now Figure 3 in the revised manuscript) as Plot 15 below. We plotted annual mean instead of percentage, and we added the WSC differences between experiments. Text has been added to describe Figure 3 (lines 202-208): "*Figure 3 shows the differences in total turbulent heat fluxes, wind stress and wind stress curl, from ECMWF_S and COARE_S with respect to NCAR. ECMWF_S wind stress is slightly weaker with respect to NCAR over the equatorial band and it is stronger elsewhere (Figure 3a). In COARE_S the wind stress is weaker than*

*NCAR over a broader region with respect to ECMWF_S, namely over the areas characterized by calm wind conditions (see Figure 1). The wind stress curl (W SC) patterns are similar for the two pairs of differences (Figure 3c), they differ only for their magnitude. As regards the QT differences (Figure 3b), a gain of heat for ECMWF_S is a clear feature over the Pacific and Atlantic equatorial regions and over EBUS with respect to NCAR."*

[Figure]

Plot 15: Annual mean differences between experiments of a) wind stress (τ) and b) total turbulent heat fluxes (QT) and c) wind stress curl (W SC) between ECMWF_S and NCAR experiments (left) and COARE_S and NCAR experiments (right). Hatching indicates significant values (95% confidence level)

**15) Then, is there a contribution to the surface cooling from an increased entrainment of cold waters in the OML (oceanic mixed layer) because of a stronger wind stress?**

We already answered this in the "Major comments" section, answer number 3.

**16) Fig9: Panel (a) is it the annual mean of the percentage variation or the percentage variation of the annual mean?**

The annual mean of the percentage variation.

**REFERENCES:**

Wang, J., Tang, D., & Sui, Y. (2010). Winter phytoplankton bloom induced by subsurface upwelling and mixed layer entrainment southwest of Luzon Strait. *Journal of Marine Systems*, *83*(3-4), 141-149.

Liang, Z., Xing, T., Wang, Y., & Zeng, L. (2019). Mixed layer heat variations in the South China Sea observed by Argo Float and reanalysis data during 2012–2015. *Sustainability*, *11*(19), 5429.

Alexander, M.A., 1992. Midlatitude atmosphere-ocean interaction during El Niño. PartI: the North Pacific Ocean. J. Climate 5, 944–958

Mendoza, V.M., Villanueva, E.E., Adem, J., 2005. On the annual cycle of the sea surface temperature and the mixed. Atmõsfera 18 (2).

---

## Referee Report (RR1)

The authors have addressed my main questions quite thoroughly, and made consequent changes to the manuscript, thank you. I have some remaining minor comments relating to clarification and language. My comments are in red.

In the following sentences, remove Fig. 3b, which shows THF. Use Fig. 3a instead. (lines 332-338)

As a consequence, the COARE_S differences in wind stress (Figure 3b)

As regard the equatorial upwelling, the weak increasing of the wind stress in the north equatorial region (e.g. northern equatorial cold front, Figure 3b) compared to NCAR wind stress (Figure 3a),

Reword: Specifically, SST and QT feedback negatively: when the SST gets anomalously cold, then QT increases, and that means that QT increases in response, the SST will tend to increase and the QT to decrease and so on. Possibly to: Specifically, SST and QT feedback negatively: when the SST gets anomalously cold, then QT increases, which will increase SST, and then QT will decrease, and so on.

Move to Code Availability statement: AeroBulk package, which is completely open source and available at https://github.com/brodeau/aerobulk (Brodeau et al., 2017).

Lines 148-155 seems to contain repeats. Can it be shortened?  Also, word "extend" is strange in this context.

Reword, use  "In contrast": *As opposed*, NCAR experiment computes THFs using bulk SST and

Add mixing? the wind stress discrepancies, due to the computation of CD and to the inclusion of the convective gustiness, may impact on the ocean dynamics by modifying the 3D ocean circulation and mixing and hence the pattern of the SST.

Add: builds up only under sunny and low wind conditions.

Regarding a response to another reviewer, in the WBCs the 1-year simulation is short due to the chaotic behavior in those regions and explains some of the noisiness in difference maps in those regions. It is worth mentioning this, if you have not already done so. It does not detract from the results of the paper.

Note the following differences are quite large, so it seems the CSWL does have a big impact on NCAR minus ECMWFG differences in the warm difference regions. The SST differences between ECMWF_NS and NCAR (Figure 6a) with respect to the SST differences between ECMWF_S and NCAR (Figure 4) present a reduction of the overall warm temperature differences

Not a time series: As it is clearly shown by the annual zonal-mean differences time-series (Fig. 7b)

Including gustiness in the ECMWF calculation produces the scalar wind differences in Figure 9a. Please clarify that differences in wind speed between ECMWF_NS and CdNC_CeEC_NS (Fig. 9a) can only be due to differences in gustiness as U does not depend on any other factors (i.e. does not depend on SST or drag coefficient).In fact it does not depend on using the ocean model so you could remove reference to experiments ECMWF_NS and CdNC_CeEC_NS for Fig. 9a and just state that it is the difference due to gustiness. In other words it is an input to the model, not an output.

Re Fig. 10a. Note that in the warm pool Equatorial regions there is some cancelling effect on wind stress of stronger U (Fig. 9a) but weaker CD (Figs. 9b,c).

Indeed, a stronger acceleration (deceleration) of southeast trades north (south) of the equator in ECMWF_NS may lead to a stronger positive (negative) curl north (south) of the Equator (Chelton et al., 2001). Note also that Chelton et al explain changes in wind stress curl due to vertical mixing in the boundary layer above SST features. So, although WSC and Ekman pumping can affect SST, it can also work in the other direction, SST affecting the WSC.

Reword "result" : which result crucial especially in wind-driven dominantly ocean regions.

in the+/-20° latitude band

(https://doi.org/10.5281/zenodo.6258085,

The reference for Levitus et al. 2013 is a bit strange. Perhaps use "Levitus, S. and coauthors"

---

## Referee Report (RR2)

**Comments to gmd-2021-436.R1**
**"The bulk parameterizations of turbulent air-sea fluxes in NEMO4: the origin of Sea Surface Temperature differences in a global model study" by Bonino et al.**

The authors have addressed all the comments raised in the first round of review. There are a couple of issues that are still unclear concerning the response to the first round of comments (listed in the minor comments) and there is one main point (major comments) that I would ask the authors to address. In addition, I strongly recommend a deep revision of the use of English, because the level is still low for a scientific publication. The list of language corrections provided in the minor comments is not exhaustive. Line numbers in this file refer to the tracked-change manuscript.
Overall, the quality of the work is good and, thus, I suggest minor revision.

**Major comments**

The presentation of the results is very hard to follow. I acknowledge that the subject is very technical and for this reason I think that a strong effort should be made in presenting the results in the clearest possible way. In particular, the figures show different metrics for different couples of experiments and the flow of the presentation goes back and forth talking about SST, tau, WSC and QT.
In the set of experiments performed, the authors nicely change one aspect at a time to go from the ECMWF_S setup to the NCAR setup. In particular, between the ECMWF_S experiment and the NCAR experiments, all aspects considered in the work (skin/SST, Cd, Ce, gustiness) are different. The logical sequence to present the results to me is: ECMWF_S, ECMWF_NS (SST instead of skin), ECMWF_NS_NG (gustiness shut off), CdNC_CeEC_NS (which I strongly suggest renaming as ECMWF_NS_NG_CdNC, in which the Cd is computed with NCAR algorithm) and NCAR (which would be equivalent to ECMWF_NS_NG_CdNC_CeNC).
First, I would check whether the annual bias of each couple of experiments (considered in the order suggested above) sum up to give the annual bias between ECMWF_S and NCAR. This could be done for the various quantities of interest (SST, tau, WSC, QT) and would explicitly show whether all these differences in the algorithms sum up linearly (at least in the annual bias). If this is the case, then, one could compute the relative importance of each correction, showing which one contributes the most in the various regions (either with a fractional variation or the correlation coefficient with respect to the full difference ECMWF_S - NCAR). For example, the annual SST bias of both couples ECMWF_S - ECMWF_NS (figure 5a) and CdNC_CeEC_NS - NCAR (figure 6c) seems to be comparable to the full bias ECMWF_S - NCAR over WBCs.
I would consider modifying the presentation of the results with two figures (one for SST and QT, and another for tau and WSC) where the maps of the following differences are shown in this order:

1. ECMWF_S - ECMWF_NS;
2. ECMWF_NS - ECMWF_NS_NG;
3. ECMWF_NS_NG - ECMWF_NS_NG_CdNC (currently named CdNC_CeEC_NS);

4. ECMWF_NS_NG_CdNC - NCAR.

In this way, the role of each modification might appear more clearly and ease the interpretation of the results.

An alternative approach to assess the importance of each difference between the algorithms would be to change one thing at a time (in some new experiments that would be ECMWF_NG, ECMWF_CdNC and ECMWF_CeNC, in addition to the ECMWF_NS experiment), compare them with ECMWF and see which one resemble the most to the NCAR-ECMWF difference. But this would require new experiments.

**Minor comments**

Add somewhere in the methods how you computed the statistical significance of the differences.

There are still many English mistakes in the text, such as (to mention some):
- L6: drives -> drive
- L44: does only partially close -> only partially closes
- L55: estimations -> estimates
- L146: on three of bulk -> on three bulk
- L164: and to lower extend to -> and, to a lower extent, to
- L170: weak -> weak-wind
- L192: two -> three
- L261: as component -> as a component
- L279: are -> is
- L283: SST differences -> SST difference
- L297: suggests that the QT drive -> suggests that QT drives

In equation (1a), according also to Brodeau et al. (2017), $u$ should be replaced with $U$ (i.e. it should be capitalized, to be consistent with the other formulae.

LL111-112: "... which may be referred to the ocean currents." -> "... which may be absolute or relative to the ocean currents."

L128: "The form of the exchange coefficients" -> "The dependence of the exchange coefficients on the wind speed"

L247: Remove "cool-skin/warm layer"

Figure 3: Especially in panels a) and b) I would suggest using a colormap that is symmetric about zero. As it is now, in fact, it enhances the negative differences, which is misleading in the interpretation of the figure.

Figure 4: Add in the caption the letters c) and d)

Figure S1: why only such a small area is selected? The pdfs over the entire WBC using high frequency data would be more informative. What is the time frequency of the data in the scatter plot? 5 days?

L312-313 - in the comparison between CdNC_CeEC_NS and ECMWF_NS there are two changes: the use of a different algorithm to compute Cd and the inclusion of gustiness in the stress computation. One should first show the effects of including the gustiness (comparison between ECMWF_NS and ECMWF_NS_NG) and then compare CdNC_CeEC_NS to ECMWF_NS_NG to show the impact of a different algorithm to compute Cd.

LL317-319: Have you tried to compute the difference between CdN and Cd? Because it looks that its pattern might also resemble the pattern of U. Thus, I would not be sure that the differences between parameterizations are only due to the different ways to compute Cd, as stated in the text.

---

## Author Response (AR2)

**REFEREE#1**

Dear Dr. Small,

we would like to thank you for the careful reading of the manuscript and the constructive comments that substantially helped to improve and clarify the paper. Answers to all your comments are detailed hereafter. Corrections to the English grammar were adopted in the revised version of the manuscript according to the reviewer's recommendations, but are not reported or discussed here. All authors agree with the modifications made to the manuscript. The comments by the referee are reported in bold followed by our response (in blue). The text added to the revised manuscript is reported in italic font. The revised manuscript that includes track changes and line numbers is provided in pdf format.

**Minor Comments**

**1) Regarding a response to another reviewer, in the WBCs the 1-year simulation is short due to the chaotic behavior in those regions and explains some of the noisiness in difference maps in those regions. It is worth mentioning this, if you have not already done so. It does not detract from the results of the paper.**

We added a sentence about it at lines 258-260:
*"It is worth mentioning that the one-year simulation might not be adequate to properly represent the mean state in WBCs regions due to the chaotic dynamics of these regions - this may explain some of the noise in the difference maps. However, this does not affect the robustness of the results."*

**2) Note the following differences are quite large, so it seems the CSWL does have a big impact on NCAR minus ECMWFG differences in the warm difference regions. The SST differences between ECMWF_NS and NCAR (Figure 6a) with respect to the SST differences between ECMWF_S and NCAR (Figure 4) present a reduction of the overall warm temperature differences.**

We added a sentence about it at lines 241-242:
"*Nevertheless, the CSWL scheme has a large impact on the positive SST difference between ECMWF_S and NCAR.*"

**3) Including gustiness in the ECMWF calculation produces the scalar wind differences in Figure 9a. Please clarify that differences in wind speed between ECMWF_NS and CdNC_CeEC_NS (Fig. 9a) can only be due to differences in gustiness as U does not depend on any other factors (i.e. does not depend on SST or drag coefficient).In fact it does not depend on using the ocean model so**

**you could remove reference to experiments ECMWF_NS and CdNC_CeEC_NS for Fig. 9a and just state that it is the difference due to gustiness. In other words it is an input to the model, not an output.**

We rephrased at lines 275-277:
*"Since U does not depend on SST and on $C_D$, including gustiness in the ECMWF calculation produces the scalar wind speed differences in Figure 9a."*

**4) Re Fig. 10a. Note that in the warm pool Equatorial regions there is some cancelling effect on wind stress of stronger U (Fig. 9a) but weaker CD (Figs. 9b,c).**

We added reference to the Equatorial warm pool at lines 294, when we discussed Figure 9.

**5) Indeed, a stronger acceleration (deceleration) of southeast trades north (south) of the equator in ECMWF_NS may lead to a stronger positive (negative) curl north (south) of the Equator (Chelton et al., 2001). Note also that Chelton et al explain changes in wind stress curl due to vertical mixing in the boundary layer above SST features. So, although WSC and Ekman pumping can affect SST, it can also work in the other direction, SST affecting the WSC.**

Thank you. In the sentence you reported here, we are just referring to stronger wind stress that leads to stronger WSC, so that we removed the reference.

**6) The reference for Levitus et al. 2013 is a bit strange. Perhaps use "Levitus, S. and coauthors**

Thank you, we corrected it.

**REFEREE #2**

Dear Referee,

we would like to thank you for the careful reading of the manuscript and the constructive comments that substantially helped to improve and clarify the paper. Answers to all your comments are detailed hereafter. Corrections to the English grammar were adopted in the revised version of the manuscript according to the reviewer's recommendations, but are not reported or discussed here. All authors agree with the modifications made to the manuscript. The comments by the referee are reported in bold followed by our response (in blue). The text added to the revised manuscript is reported in italic font. The revised manuscript that includes track changes and line numbers is provided in pdf format.

In the following answers, we use 'Figure' to identify the figures in the updated manuscript and we use 'Plot' to identify the figures in this document.

**Comments**

**1) The authors have addressed all the comments raised in the first round of review. There are a couple of issues that are still unclear concerning the response to the first round of comments (listed in the minor comments) and there is one main point (major comments) that I would ask the authors to address. In addition, I strongly recommend a deep revision of the use of English, because the level is still low for a scientific publication.**

The manuscript has been revised for the use of English by a company specialized in English for academics (https://e4ac.com/). The revision is certified.

**2) The presentation of the results is very hard to follow. I acknowledge that the subject is very technical and for this reason I think that a strong effort should be made in presenting the results in the clearest possible way. In particular, the figures show different metrics for different couples of experiments and the flow of the presentation goes back and forth talking about SST, tau, WSC and QT. In the set of experiments performed, the authors nicely change one aspect at a time to go from the ECMWF_S setup to the NCAR setup. In particular, between the ECMWF_S experiment and the NCAR experiments, all aspects considered in the work (skin/SST, Cd, Ce, gustiness) are different. The logical sequence to present the results to me is: ECMWF_S, ECMWF_NS (SST instead of skin), ECMWF_NS_NG (gustiness shut off), CdNC_CeEC_NS (which I strongly suggest renaming as ECMWF_NS_NG_CdNC, in which the Cd is computed with NCAR algorithm) and NCAR (which would be equivalent to**

ECMWF_NS_NG_CdNC_CeNC). First, I would check whether the annual bias of each couple of experiments (considered in the order suggested above) sum up to give the annual bias between ECMWF_S and NCAR. This could be done for the various quantities of interest (SST, tau, WSC, QT) and would explicitly show whether all these differences in the algorithms sum up linearly (at least in the annual bias). If this is the case, then, one could compute the relative importance of each correction, showing which one contributes the most in the various regions (either with a fractional variation or the correlation coefficient with respect to the full difference ECMWF_S - NCAR). For example, the annual SST bias of both couples ECMWF_S - ECMWF_NS (figure 5a) and CdNC_CeEC_NS - NCAR (figure 6c) seems to be comparable to the full bias ECMWF_S - NCAR over WBCs. I would consider modifying the presentation of the results with two figures (one for SST and QT, and another for tau and WSC) where the maps of the following differences are shown in this order:

1. ECMWF_S - ECMWF_NS;
2. ECMWF_NS - ECMWF_NS_NG;
3. ECMWF_NS_NG - ECMWF_NS_NG_CdNC (currently named CdNC_CeEC_NS);
4. ECMWF_NS_NG_CdNC - NCAR.

In this way, the role of each modification might appear more clearly and ease the interpretation of the results.

An alternative approach to assess the importance of each difference between the algorithms would be to change one thing at a time (in some new experiments that would be ECMWF_NG, ECMWF_CdNC and ECMWF_CeNC, in addition to the ECMWF_NS experiment), compare them with ECMWF and see which one resemble the most to the NCAR-ECMWF difference. But this would require new experiments.

Thanks for the suggestions, we computed all the TASFs (i.e. $\tau$ and QT) and WSC annual mean differences for each pair of experiments. The annual differences not discussed in the manuscript are reported in the supplementary material. In particular, we added Plot 2 (Annual mean differences of wind stress, and wind stress curl between ECMWF_S and ECMWF_NS) as Figure S1 for ECMWF_S - ECMWF_NS (section 3.2), and Plot 3 (Annual mean differences of wind stress, and wind stress curl between ECMWF_S and ECMWF_NS) as Figure S3 for CdNC_CeEC_NS - NCAR (section 3.3). We also checked that the annual bias of each pair of experiments sums up to give the annual bias between ECMWF_S and NCAR. This is verified for all the variables. We added a sentence on this at lines 221-223 in Section 3.1: "*It is worth mentioning that the annual mean differences (i.e. SST, $\tau$, WSC and QT) of each pair of experiments discussed in the following sections sum up linearly to give the annual mean difference between ECMWF_S and NCAR (not shown).*"

Nevertheless, we decided to not include the ECMWF_NS - ECMWF_NS_NG as a separate couple of experiments. The wind stress, $C_D$ and SST differences between ECMWF_NS - ECMWF_NS_NG (Plot 1) are not relevant enough (less than 10% of

the difference ECMWF_S - NCAR, Figure 3) to have a dedicated paragraph in the manuscript.

The couples of experiments in the manuscript are still:

1. ECMWF_S - ECMWF_NS (section 3.2);
2. CdNC_CeEC_NS - NCAR (section 3.3);
3. ECMWF_NS - CdNC_CeEC_NS (section 3.4).

However, the difference in wind stress between ECMWF_NS and ECMWF_NS_NG is still discussed in section 3.4 and shown in Figure S2.

Moreover, the name ECMWF_NS_NG_CdNC seems to refer just to ECMWF parameterization, when, actually, the used bulk algorithm is a mixture of ECMWF and NCAR parameterizations. We therefore prefer to retain the name CdNC_CeEC_NS.

We clarified our logical flow at lines 177-187:

"*Here we discuss the parameterization-related discrepancies in the control experiments in terms of TASFs (i.e. $\tau$ and QT), WSC, SST and meridional heat transport (section 3.1). Then, we try to analyze the contribution of various aspects of the parameterizations in driving these SST and meridional transport discrepancies. In particular, the comparison between ECMWF_S and ECMWF_NS is used to determine the skin temperature contribution (section 3.2), while the comparisons between CdNC_CeEC_NS and NCAR (section 3.3) and between CdNC_CeEC_NS and ECMWF_NS (section 3.4) teach us about the Bulk Transfer Coefficients contribution. In section 3.4, we also compare ECMWF_NS_NG and ECMWF_NS experiments to show the effect of the inclusion of convective gustiness in the wind speed calculation on wind stress computation (shown in the supplementary material). For each couple of experiments, we only show the differences in TASFs and their components (e.g. U, $C_D$, $C_E$) which are relevant to understand the SST or meridional heat transport discrepancies. The complementary TASFs differences are reported in the supplementary material. We analyze annual mean differences between experiments and assess their statistical significance using t-test.*"

[Figure]

Plot 1: Annual mean differences of a) wind stress (τ), b) Sea Surface Temperature (SST) and c) wind stress transfer coefficient (CD) between ECMWF_NS and ECMWF_NS_NG. Hatching indicates significant values (95% confidence level)

[Figure]

Plot 2: Annual mean differences of a) wind stress ($\tau$), and b) wind stress curl (WSC) between ECMWF_S and ECMWF_NS. Hatching indicates significant values (95% confidence level)

[Figure]

Plot 3: Annual mean differences of a) wind stress (τ), and b) wind stress curl (WSC) between CdNC_CeEC_NS and NCAR. Hatching indicates significant values (95% confidence level)

**3) Add somewhere in the methods how you computed the statistical significance of the differences.**

Thank you, we added a sentence on statistical significance at line 186-187: *"We analyze annual mean differences between experiments and assess their statistical significance using t-test."*

**4) In equation (1a), according also to Brodeau et al. (2017), u should be replaced with U (i.e. it should be capitalized, to be consistent with the other formulae.**

Thank you, we corrected it.

**5) Figure 3: Especially in panels a) and b) I would suggest using a colormap that is symmetric about zero. As it is now, in fact, it enhances the negative differences, which is misleading in the interpretation of the figure.**

Thank you for the suggestion. With the symmetric colorbar (Plot 4), we cannot appreciate the τ difference between the two couples of experiments in the equatorial

band, an important feature used to discuss COARE_S - NCAR in section 3.4. Therefore we decided to retain the Figures with the non symmetric colorbar.

[Figure]

Plot 4: Annual mean differences between experiments of a) wind stress (τ) between ECMWF_S and NCAR experiments (top panel) and COARE_S and NCAR experiments (bottom panel). Hatching indicates significant values (95% confidence level)

**6) Figure S1: why only such a small area is selected? The pdfs over the entire WBC using high frequency data would be more informative. What is the time frequency of the data in the scatter plot? 5 days?**

The analysis was performed on a set of different areas in the WBC regions. Results are all consistent. We specified that the plot shows the relationship between total turbulent fluxes and the air-sea virtual temperature difference for selected grid points in one of the boxes as a matter of clearness. Yes, 5 days is the frequency of the data.

**7) LL317-319: Have you tried to compute the difference between CdN and Cd? Because it looks that its pattern might also resemble the pattern of U. Thus, I would not be sure that the differences between parameterizations are only due to the different ways to compute Cd, as stated in the text.**

In lines 317-319, since the pattern differences in Figure 9b and Figure 9c are very similar, we are suggesting that the differences in $C_D$ between parameterizations are related to the computation of neutral coefficient ($C_{DN}$) calculation rather than to its

stability correction (term to add to $C_{DN}$ to get $C_D$ coefficients). Plot 5 shows the difference between $C_D$ and $C_{DN}$ for (a) ECMWF, (b) NCAR and c) shows a-b. The patterns of $C_D$ - $C_{DN}$ differences are really similar between experiments. The stability correction of NCAR is stronger than the stability correction of ECMWF everywhere. Plot 5c basically shows the contribution of the atmospheric stability function to the $C_D$ differences pattern (Figure 9b). Even though stability has for sure its role in determining the $C_D$ values, the pattern of $C_D$ differences between experiments (positive and negative areas) are related to the different $C_{DN}$ computation.

[Figure]

Plot 5: Annual mean differences of CD - CDN for a) ECMWF_S and b) CdNC_CeEC_NS; c) differences between a) and b).

**8) L312-313 - in the comparison between CdNC_CeEC_NS and ECMWF_NS there are two changes: the use of a different algorithm to compute Cd and the inclusion of gustiness in the stress computation. One should first show the effects of including the gustiness (comparison between ECMWF_NS and ECMWF_NS_NG) and then compare CdNC_CeEC_NS to ECMWF_NS_NG to show the impact of a different algorithm to compute Cd.**

See answer to comment 2 above.